# Temporal fluxomics reveals oscillations in TCA cycle flux throughout the mammalian cell cycle

Eunyong Ahn[1,†], Praveen Kumar[2,†], Dzmitry Mukha[2], Amit Tzur[3,4] & Tomer Shlomi[1,2,5,*] (iD)

## Abstract

Cellular metabolic demands change throughout the cell cycle. Nevertheless, a characterization of how metabolic fluxes adapt to the changing demands throughout the cell cycle is lacking. Here, we developed a temporal-fluxomics approach to derive a comprehensive and quantitative view of alterations in metabolic fluxes throughout the mammalian cell cycle. This is achieved by combining pulse-chase LC-MS-based isotope tracing in synchronized cell populations with computational deconvolution and metabolic flux modeling. We find that TCA cycle fluxes are rewired as cells progress through the cell cycle with complementary oscillations of glucose versus glutamine-derived fluxes: Oxidation of glucose-derived flux peaks in late G1 phase, while oxidative and reductive glutamine metabolism dominates S phase. These complementary flux oscillations maintain a constant production rate of reducing equivalents and oxidative phosphorylation flux throughout the cell cycle. The shift from glucose to glutamine oxidation in S phase plays an important role in cell cycle progression and cell proliferation.

**Keywords** cell cycle; cellular metabolism; isotope tracing; LC-MS; metabolic flux analysis

**Subject Categories** Cell Cycle; Genome-Scale & Integrative Biology; Metabolism

**Mol Syst Biol. (2017) 13: 953**

## Introduction

Cell cycle progression is tightly interlinked with cellular metabolism (Kaplon *et al*, 2015). The availability of sufficient metabolic nutrients and intracellular energy status controls the ability of cells to enter and progress through cell cycle. The absence of glucose was first shown to arrest cells at the G1/S restriction point (Blagosklonny & Pardee, 2002). More recently, cellular energy status (ATP/AMP ratio) was found to regulate canonical cell cycle signaling pathways via AMP-activated protein kinase (AMPK; Banko *et al*, 2011). The mammalian target of rapamycin (mTOR) plays a central role in regulating cell cycle progression and growth, integrating stimuli of amino acid, energy, and oxygen availability (Fingar & Blenis, 2004; Cuyàs *et al*, 2014). Cell cycle progression is further controlled by intracellular metabolites affecting epigenetics: Nuclear acetyl-CoA levels, determined by nuclear ATP citrate lyase (ACL; Wellen *et al*, 2009) and pyruvate dehydrogenase (PDH; Sutendra *et al*, 2014), regulate the acetylation of histones and thus control cell cycle progression (Berger, 2007; Li *et al*, 2007). Additionally, several metabolic enzymes were shown to directly regulate the cell cycle machinery, including PFKFB3 and PKM2, controlling the activity of cyclins and cyclin-dependent kinase (CDK) inhibitors in the nucleus (Yalcin *et al*, 2009; Yang *et al*, 2011, 2012).

Signaling pathways that coordinate cell cycle progression further regulate metabolic activity to support the changing metabolic demands throughout the cell cycle. The ubiquitin proteasome system, which tightly controls the concentration of cyclins, regulates the activity of two key enzymes in glucose and glutamine metabolism (Almeida *et al*, 2010; Tudzarova *et al*, 2011; Estevez-Garcia *et al*, 2014); the ubiquitin ligase anaphase-promoting complex/cyclosome (APC/C) and ligase Skp1/cullin/F-box protein (SCF) complex control glycolytic flux via PFKFB3, restricting its expression to late G1 and early S; APC/C also regulates glutaminolysis via glutaminase 1 (GLS1), whose expression is induced in S and G2/M. Cyclins and cyclin-CDK complexes were further suggested to regulate central metabolic activities, including glycolysis, lipogenesis, and mitochondrial activity (Hsieh *et al*, 2008; Bienvenu *et al*, 2010). Furthermore, central oncogenes and tumor suppressors that control proliferation, growth, and cell cycle can stimulate the expression of enzymes that mediate glycolysis and glutaminolysis (Levine & Puzio-Kuter, 2010).

While the cell cycle machinery was found to regulate the concentration of key metabolic enzymes, an understanding of how the actual rate of metabolic reactions and pathway (i.e., metabolic flux) changes throughout the cell cycle is still fundamentally missing. Metabolic flux is a not a directly measurable quantity and is typically inferred using isotope tracing techniques coupled with computational metabolic flux analysis (MFA; Wiechert, 2002; Sauer, 2006).

---

1 Department of Computer Science, Technion, Haifa, Israel
2 Department of Biology, Technion, Haifa, Israel
3 Faculty of Life Sciences, Bar-Ilan University, Ramat Gan, Israel
4 The Institute of Nanotechnology and Advanced Materials, Bar-Ilan University, Ramat Gan, Israel
5 Lokey Center for Life Science and Engineering, Technion, Haifa, Israel
 *Corresponding author. Tel: +972 77 887 1543; E-mail: tomersh@cs.technion.ac.il
 †These authors contributed equally to this work

Isotope tracing coupled with MFA is commonly used to address problems in biotechnology and medicine and has recently become a central technique in studies of cancer cellular metabolism (Metallo *et al*, 2009; Duckwall *et al*, 2013). Applied to cell populations with cells at different phases of the cell cycle, this approach typically estimates the average flux throughout the cell cycle.

Here, we present a temporal-fluxomics approach for quantifying cell cycle-dependent oscillations in metabolic flux, combining isotope tracing in synchronized cell populations with computational deconvolution and metabolic network modeling. We applied this approach to derive a first comprehensive and quantitative view of flux dynamics in central metabolism of proliferating cancer cells. The analysis adds a temporal dimension to our understanding of TCA cycle metabolism, showing major oscillations in the oxidation/reduction of glucose versus glutamine-derived fluxes as cells progress through the cell cycle.

## Results

### Cellular concentration of central metabolic intermediates oscillate throughout the cell cycle

To study metabolic dynamics throughout cell cycle, we synchronized HeLa cells using double thymidine block and applied high-throughput LC-MS-based targeted metabolomics analysis to synchronized cell populations ($> 10^6$ cells per sample) in 3-h intervals for two complete cell cycles (see cell synchronization dynamics measured via propodium iodide staining/FACS analysis in Fig 1A; Appendix Fig S1; Materials and Methods). To obtain a reliable and accurate view of periodic metabolic oscillations and to overcome a potential perturbation of metabolism due to synchronization-induced growth arrest, we let synchronized cells complete one cell cycle before starting the LC-MS analysis (9 h after cells are released in G1/S). Measured metabolite abundances in the synchronized cells were normalized by total cell volume in each time point to determine metabolite concentrations.

As cell synchronization is gradually lost with time due to inherent non-genetic cell-to-cell variability (a phenomenon also known as "dispersion"), the distribution of cell cycle phases in the synchronized cell population becomes similar to that of non-synchronized cells after completing three rounds of replications (Fig 1A). To account for the loss of synchrony and to precisely quantify oscillations in metabolite levels, we employed "computational synchronization" (Bar-Joseph *et al*, 2008): We constructed a probabilistic model that describes the dynamics of the cell population loosing synchrony, assuming that each cell has its own "internal clock" which controls the cell cycle progression rate (see Synchronization loss model in Materials and Methods). The parameters of the model were estimated by fitting a simulation of how the synchronized cell population progresses through the different phases of the cell cycle with corresponding FACS measurements, finding that cell–cell variability in the rate of cell cycle progression through the cell cycle is 11% (Fig 1B, Materials and Methods). We used this model for *computational deconvolution* of measured metabolomics data, estimating metabolite concentration dynamics throughout the cell cycle, circumventing the impact of cell dispersion (Materials and Methods). Inferring the dynamics of metabolite concentrations

throughout the cell cycle (rather than that of metabolite abundances) further required estimates of the dynamics of cell volume throughout the cell cycle. The latter was estimated based on deconvolution of total cell volume measurements performed in the synchronized cell population (Materials and Methods, Fig 1C). For example, the concentration of the nucleotide cytidine triphosphate (CTP) was found to oscillate throughout cell cycle, showing a ~50% increase in concentration in G1 phase versus G2/M phase (measured and deconvoluted concentrations shown in Fig 1D). As shown, the magnitude of the oscillation drops with time and converges to the steady-state concentration measured in non-synchronized cells.

Our analysis reveals 57 metabolites whose concentrations significantly oscillate throughout the cell cycle (Fig 2, Dataset EV1, FDR-corrected *P*-value < 0.05, Materials and Methods). Oscillations in nearly 44% of these metabolites could be detected only when *in silico* synchronization via computational deconvolution was applied (i.e., equation 7 in Materials and Methods), emphasizing the strength of our pipeline. The median size of the observed oscillations is ~60% (difference between maximal and minimal concentration throughout the cell cycle); roughly one-quarter of these metabolites show concentration changes larger than twofold throughout cell cycle. A significantly high fraction of the metabolites peaks either in late G1 (~50% in the second half of G1, *P*-value < $10^{-5}$, compared with the expected fraction assuming that concentration peaks are uniformly distributed throughout the entire cell cycle) or early S (~35% in the first half of S, *P*-value < 0.002).

The oscillating metabolites include glycolytic and TCA cycle intermediates, nucleotides, amino acids, and energy and redox cofactors. Expectedly, the concentrations of the deoxynucleotides dCTP and dATP increase in S phase, when utilized for DNA replication. Intermediates in polyamine metabolism (5-methylthioadenosine, acetyl-putrescine, S-adenosyl-L-methionine, and S-adenosyl-L-homocysteine) show marked oscillations, in accordance with the known cell cycle-dependent activity of this pathway (Oredsson, 2003). Several glycolytic metabolites peak during G1/S transition, in accordance with reports of increased glycolytic flux at this cell cycle phase (Colombo *et al*, 2011; Tudzarova *et al*, 2011). Cellular ATP/ADP ratio and redox potential (NADH/NAD$^+$) further show a ~50% increase in the G1/S transition (Appendix Fig S2). Intracellular concentration of non-essential amino acids synthesized from consumed glutamine, including glutamate, ornithine, proline, and aspartate peak in S phase, in accordance with a reported increase in glutamine dependence in S phase (Gaglio *et al*, 2009; Colombo *et al*, 2011). Intriguingly, we find that different TCA cycle metabolites peak in distinct cell cycle phases: Acetyl-CoA and citrate peak in G1/S, while malic acid and α-ketoglutarate peak in late S, suggesting that the TCA cycle is rewired as cells progress through the cell cycle.

### Time-resolved fluxomics reveals increased glycolytic flux into TCA cycle in G1/S transition

To observe metabolic flux dynamics in TCA cycle and in branching pathways throughout the cell cycle, we performed pulse-chase isotopic tracing experiments in synchronized HeLa cells with [U-$^{13}$C]-glucose and [U-$^{13}$C]-glutamine (1-h feeding), every 3 h for two cell cycles (Fig 3A and B, Materials and Methods). Here, LC-MS was utilized to measure the mass-isotopomer distribution of metabolites (i.e., the fraction of each metabolite pool having zero,

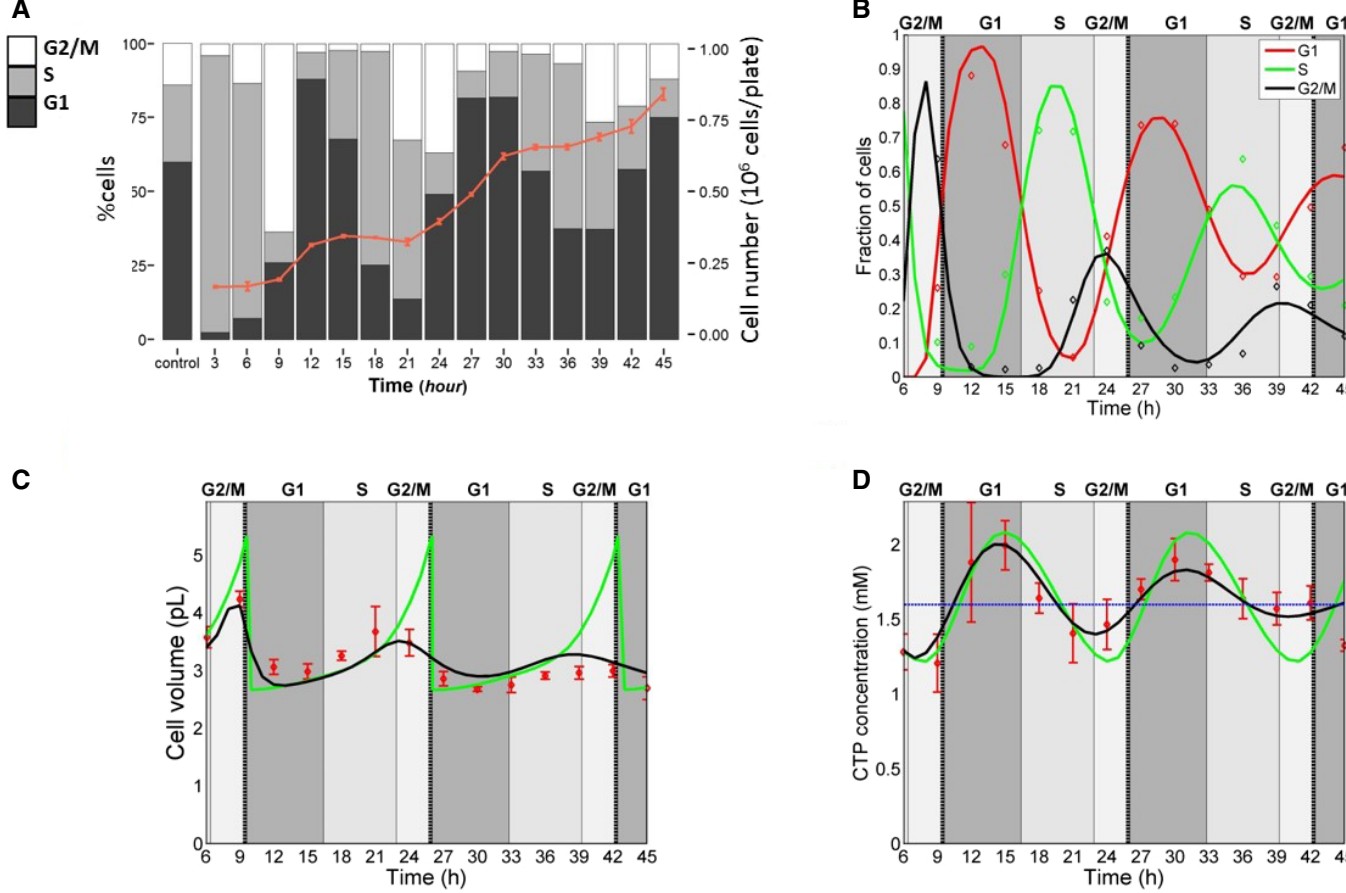

**Figure 1. Detection of oscillations in metabolite concentrations throughout the cell cycle in HeLa cells revealed by LC-MS-based metabolomics of synchronized HeLa cells and computational deconvolution.**

A  Synchronization dynamics of a population of HeLa cells within almost three complete cell cycles measured via PI staining followed by FACS analysis. The increase in cell number following each mitosis is shown by an overlaid curve (in orange, mean and s.d. of $n = 3$).

B  Computational modeling of the synchronization loss, considering 11% cell–cell variation in doubling time, shows that the simulated fraction of the cells in G1, S, and G2/M in the synchronized cells throughout the cell cycle (straight lines) match experimental measurements (marked with asterisk).

C  The measured average cell volume in the synchronized cell population (red, mean and s.d. of $n = 3$, $v(t)$ in equation 6), the deconvoluted signal (in case of no synchronization loss, green, $v'(x)$ in equation 6), and the simulated average cell volume considering the loss in synchronization (black, matching the measured concentration data, equation 6).

D  The measured concentration of CTP in synchronized cells shown in red (mean and s.d. of $n = 5$, $u_i(t)$ in equation 7), the deconvoluted concentration dynamics, in case of no synchronization loss (green, $u'(x)$ in equation 7), and the expected concentration dynamics based on the deconvoluted concentrations and considering the loss in synchronization, matching the measured concentrations (black, equation 7). The measured concentrations converge toward the steady-state concentrations measured in non-synchronized cells (horizontal blue line).

one, and two labeled carbon atoms) after 1 h of feeding with the isotopic tracers. Computational deconvolution was employed to analyze oscillations in metabolite isotopic labeling patterns while considering cell dispersion. The deconvolution approach is based on the observation that the measured fractional isotopic labeling of a metabolite in the synchronized cell population represents the average labeling in cells with distinct intrinsic times, weighted by the metabolite pool size in these cells (i.e., the measured isotopic labeling pattern is biased toward that of cells with an intrinsic time in which the metabolite pool size is larger than in others, Materials and Methods). Overall, we detected statistically significant oscillations in the isotopic labeling pattern of 21 metabolites when feeding isotopic glucose, and 16 metabolites when feeding isotopic glutamine (FDR-corrected $P$-value < 0.05, Figs 3 and 4, Dataset EV2).

The inferred oscillations in metabolite isotopic labeling and concentrations were used to computationally analyze metabolic flux dynamics throughout the cell cycle, utilizing a variant of kinetic flux profiling (KFP; Yuan *et al*, 2008; Fig 5, in units of nmole/μl-cells/h, i.e., mM/h, Materials and Methods). Specifically, given a metabolite whose isotopic labeling dynamics throughout the cell cycle was inferred as explained above, we search for the most likely transient production and consumption fluxes in each 1-h interval through the cell cycle, such that the simulated labeling kinetics of this metabolite (within the 1-h interval) would optimally match the experimental measurements (Materials and Methods, Appendix Figs S3–S8). The simulation of the isotopic labeling kinetics of a metabolite of interest within a 1-h time interval is performed via an ordinary differential equations (ODE) model,

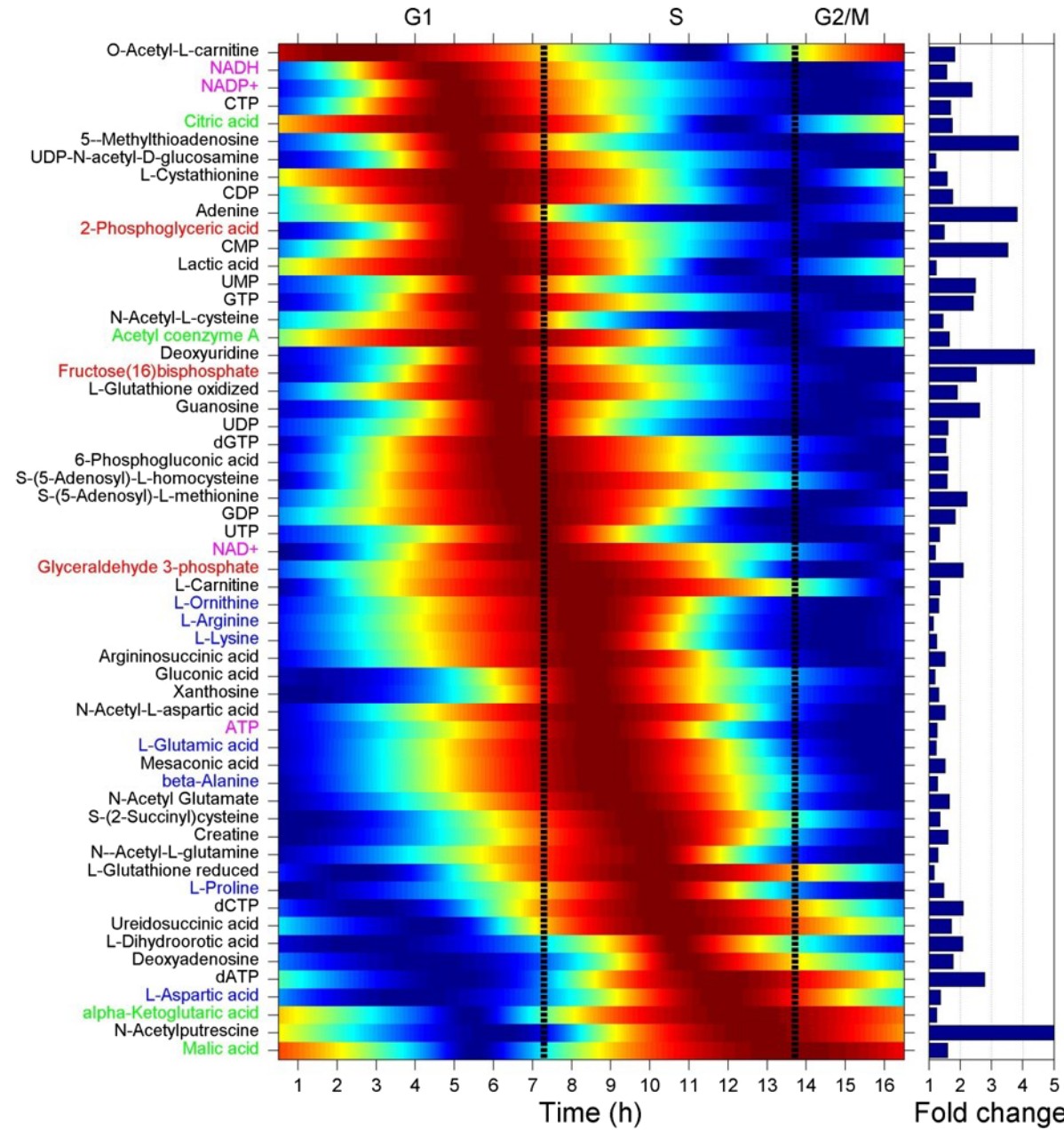

**Figure 2. Oscillation in metabolite concentrations throughout the cell cycle in HeLa cells.**

The figure shows metabolites found to significantly oscillate throughout the cell cycle (concentrations normalized per metabolite, maximal concentration in red, minimal concentration in blue). The amplitude of the oscillations is shown on the right. Metabolites are color-coded according to metabolic pathways: energy/redox cofactors (purple), amino acids (blue), glycolytic metabolites (red), and TCA cycle metabolites (green).

relying on the inferred concentration of the metabolite within this time interval (considering that a metabolite with a larger pool size would take more time to label, per unit of flux), as well as the isotopic labeling kinetics of intermediates that produce this metabolite. While KFP is typically applied to estimate fluxes under metabolic steady state (in which fluxes satisfy a stoichiometric mass-balance constraint), here, we constrain the difference between transient fluxes that produce and consume a certain metabolite according to the measured momentary change in the concentration of that metabolite.

Oscillations in the isotopic labeling pattern of TCA cycle intermediates when feeding isotopic glucose suggest that glucose-derived flux into TCA cycle increases in G1 phase and then drops in S phase. The fractional labeling of the m + 2 form of the TCA cycle intermediates citrate, α-ketoglutarate, and malate drops in S phase (Fig 3C–E). Feeding cells with isotopic glutamine, we further observed a drop in citrate m + 4 produced from oxaloacetate via citrate synthase in S phase (Fig 3F). Combined with the drop in citrate concentration during S phase (Fig 3G), metabolic modeling reveals a ~2-fold decrease in glycolytic flux into TCA cycle as cells progress through S phase;

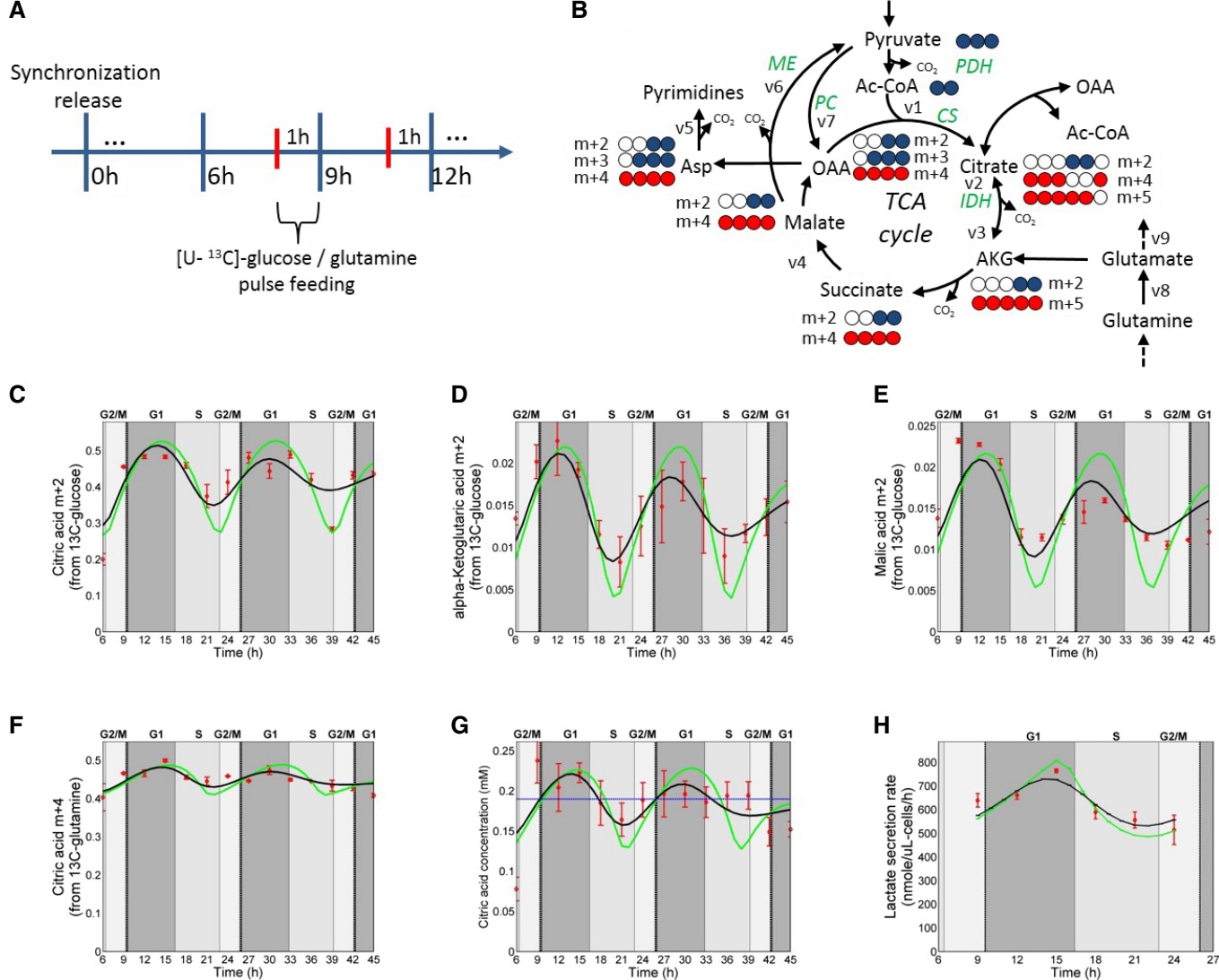

**Figure 3.  Oscillations in isotopic labeling of TCA cycle metabolites throughout the cell cycle from [U-$^{13}$C]-glucose show induced glycolytic flux into TCA cycle in G1/S.**

A    Experimental scheme for a series of pulse-chase isotope tracing experiments in synchronized cells.

B    Atom tracing of TCA cycle metabolites from [U-$^{13}$C]-glucose (blue) and [U-$^{13}$C]-glutamine (red).

C–E    Measured relative fraction of the m + 2 labeling of TCA cycle intermediates after feeding [U-$^{13}$C]-glucose (red, mean and s.d. of $n$ = 3), the deconvoluted signal (green), and the expected labeling dynamics considering the loss in synchronization (black, representing TCA cycle oxidation of glucose-derived acetyl-CoA).

F    Oscillations in citrate m + 4 labeling after feeding [U-$^{13}$C]-glutamine (mean and s.d. of $n$ = 3; experimentally measured labeling dynamics shown in red, the deconvoluted signal in green, and the expected labeling dynamics considering the synchronization loss in black).

G    Oscillations in the total citrate concentration throughout the cell cycle (mean and s.d. of $n$ = 5; experimentally measured concentration dynamics shown in red, the deconvoluted signal in green, the expected concentration dynamics considering the synchronization loss in black, and the measured concentration in non-synchronized cells in blue).

H    The measured lactate secretion flux in synchronized cells shown in red (mean and s.d. of $n$ = 5, $f_i(t)$ in equation 9), the deconvoluted secretion flux dynamics, in case of no synchronization loss (green, $f_i'(x)$ in equation 10), and the expected secretion flux based on the deconvoluted fluxes and considering the loss in synchronization, matching the measured fluxes (black, equation 10).

citrate synthase flux drops from ~6 mM/h in G1/S phase to ~3 mM/h in late S phase (Fig 5A, Appendix Fig S3). TCA cycle oxidation of citrate via isocitrate dehydrogenase (IDH) shows similar flux dynamics, with ~2-fold drop in S phase (Fig 5C, Appendix Fig S4).

To examine whether the increase in glucose-derived flux into TCA cycle in G1/S is associated with increased in glycolytic flux,

we measured lactate concentrations in the culture media in the synchronized cell population followed by computational deconvolution (Fig 3H, Materials and Methods). We find a ~65% increase in lactate secretion rate in G1/S transition. Considering that the average lactate secretion rate throughout the cell cycle is two orders of magnitude higher than that of pathways that

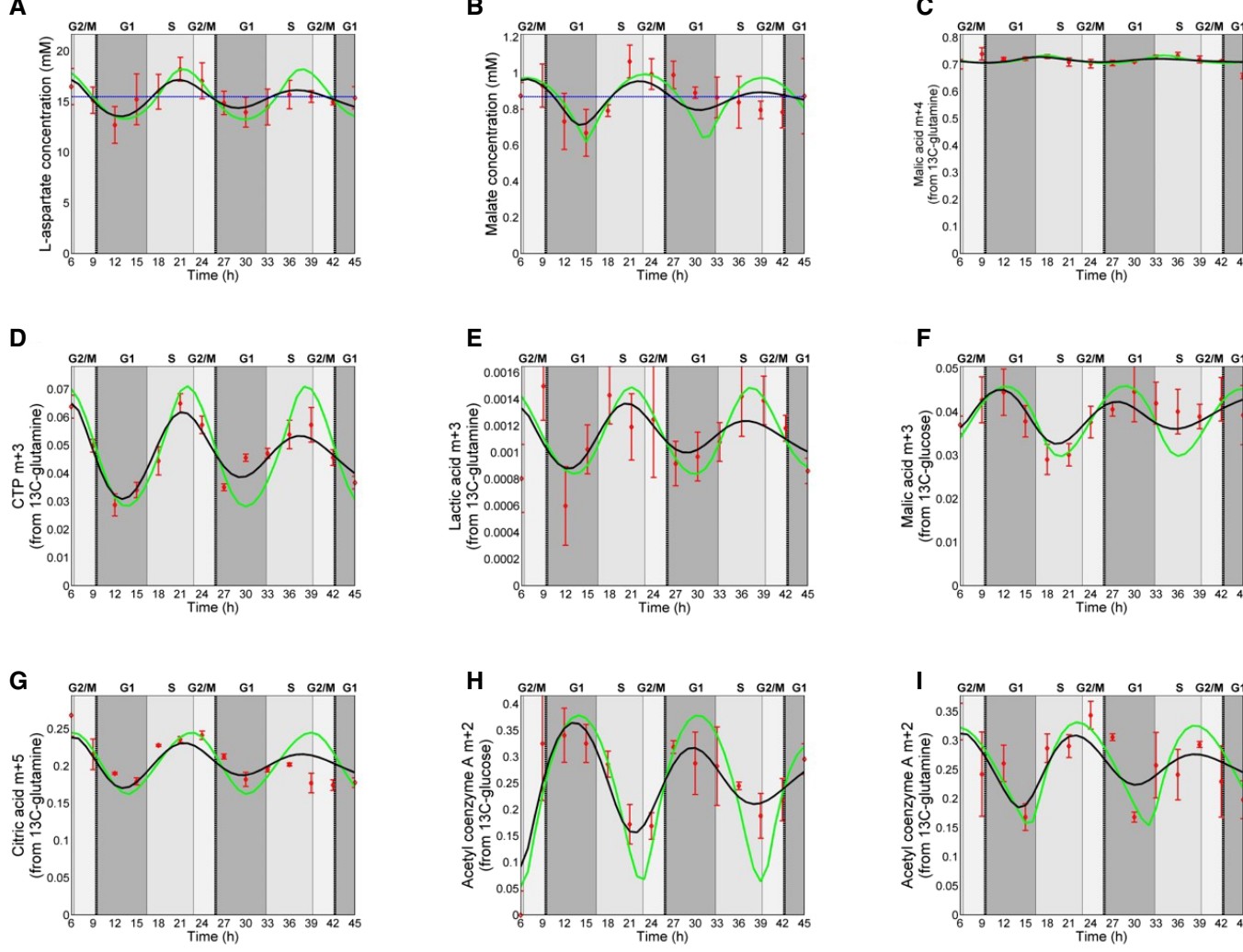

**Figure 4. Oscillations in isotopic labeling of TCA cycle metabolites throughout the cell cycle from [U-¹³C]-glutamine show induced oxidative and reductive glutamine metabolism in S phase.**

A, B   Oscillations in aspartate (A) and malate (B) concentrations throughout the cell cycle when feeding [U-¹³C]-glutamine (representing oxidative TCA cycle activity).

C   Uniform malate m + 4 labeling throughout the cell cycle (combined with the increase in malate concentration in S phase representing increased oxidative TCA cycle flux in S phase).

D   Oscillations in pyrimidines m + 3 labeling throughout the cell cycle when feeding [U-¹³C]-glutamine (representing *de novo* pyrimidine biosynthesis).

E   Oscillations in lactate m + 3 labeling throughout the cell cycle when feeding [U-¹³C]-glutamine (representing malic enzyme activity).

F   Oscillations in malate m + 3 throughout the cell cycle when feeding [U-¹³C]-glucose (representing pyruvate carboxylase activity).

G   Oscillations in citrate m + 5 when feeding [U-¹³C]-glutamine throughout the cell cycle (representing reductive IDH flux).

H   Oscillations in acetyl-CoA m + 2 when feeding [U-¹³C]-glucose, representing oxidative glucose metabolism.

I   Oscillations in acetyl-CoA m + 2 when feeding [U-¹³C]-glutamine, representing reductive glutamine metabolism.

Data information: (A, B) Measured metabolite concentrations are shown in red (showing mean and s.d. of $n = 5$), the deconvoluted signal in green, the expected concentration dynamics considering the synchronization loss in black, and the measured concentrations in non-synchronized cells in blue. (C–I) Measured fractional isotopic labeling are shown in red (showing mean and s.d. of $n = 3$), the deconvoluted signal in green, and the expected isotopic labeling dynamics considering the synchronization loss in black.

branch out from glycolysis, the observed oscillation in lactate secretion represents cell cycle-dependent changes in glycolytic flux: The average lactate secretion throughout the cell cycle is ~600 mM/h, while oxidative pentose-phosphate pathway (PPP) is ~3 mM/h, reductive PPP is ~3 mM/h, glycogenesis is ~0.3 mM/h, and serine biosynthesis is below 1 mM/h (Appendix Fig S8). Overall, our data show that the increase in glycolytic flux in G1/S phase co-occurs with the increased glucose-driven flux entering the TCA cycle. Notably, analyzing oscillations in glycolytic flux based on direct measurement of changes in glucose consumption throughout the cell cycle (rather than based on lactate secretion) was not possible due to technical difficulty in accurately quantifying glucose consumption by synchronized cells within 3-h time intervals (considering that the synchronized cell population consumes ~1% of the glucose in media within this short time period).

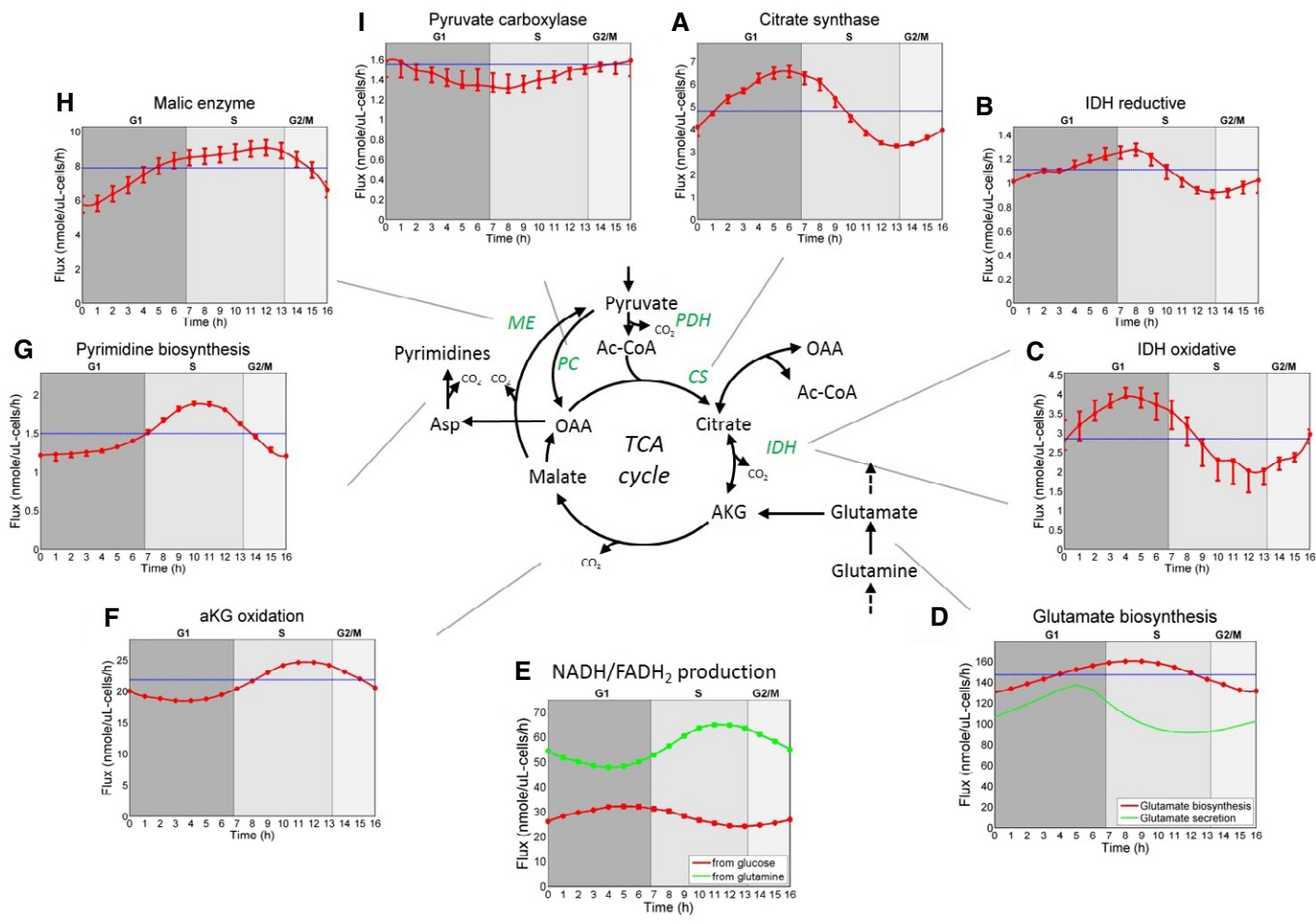

**Figure 5.  Complementary oscillations of glucose versus glutamine-derived fluxes in TCA cycle.**

A–I   Oscillations in metabolic flux throughout the cell cycle (in mM/h), computed based on metabolic modeling of measured oscillations in metabolite concentrations and isotopic labeling (red and green marks represent optimal estimates of transient flux with 95% CI). Blue lines represent average fluxes inferred in a non-synchronized cell population. As shown, glucose-derived flux into TCA cycle peaks in late G1 phase, while oxidative and reductive glutamine metabolism dominates S phase.

## Induced oxidative and reductive glutamine metabolism compensates for the decreased glycolytic flux into TCA cycle in S phase

Glutamine feeds TCA cycle flux by producing glutamate, which is converted to α-ketoglutarate either via transamination or by glutamate dehydrogenase. As a first estimation of the cell cycle dynamics of glutamine-derived flux into the TCA cycle, we quantified cell cycle-dependent glutamate production from glutamine versus glutamate secretion to the media. Tracing the m + 5 labeling dynamics of glutamate when feeding [U-$^{13}$C]-glutamine and glutamate concentration throughout the cell cycle suggests that glutamate production flux increases by 25% in S phase compared to G1 (Fig 5D). This is evident by a marked increase in glutamate concentration in S phase and similar m + 5 glutamate labeling kinetics throughout the cell cycle (Appendix Fig S4). Glutamate secretion rate to the culture medium shows a marked drop in S phase, suggesting increased availability of glutamate for feeding the TCA cycle flux in S phase (Fig 5D). The increased entry of glutamine-derived flux into the

TCA cycle in S phase is followed by a ~40% increase in α-ketoglutarate oxidation (Fig 5F, Appendix Fig S5). This is evident by the marked increase in malate and aspartate concentration in S phase (Fig 4A and B) and barely altered m + 4 and m + 3 labeling kinetics of these metabolites throughout the cell cycle, respectively (Fig 4C and Appendix Fig S9).

The increased glutamine-derived anaplerotic flux into the TCA cycle in S phase (via net production of the TCA cycle intermediate α-ketoglutarate) is balanced by oscillations in cataplerotic fluxes, consuming TCA cycle intermediates for biosynthetic and bioenergetics purposes: We find a ~70% increase in pyrimidine biosynthesis flux in S phase, consuming oxaloacetate from TCA cycle (transaminated to produce aspartate, Fig 5G). This is evident by a marked increase in the m + 3 labeling of pyrimidines in S phase (Fig 4D and Appendix Fig S10), while considering the oscillations in the labeling kinetics of carbamoyl-aspartate in pyrimidine biosynthesis and pyrimidine concentrations (Appendix Fig S6). Oscillations in the biosynthetic flux of pyrimidines as well as purines is further supported by an increased m + 5 labeling of pyrimidines and

purines in S phase (i.e., having all five ribose carbons labeled) upon feeding with isotopic glucose (Dataset EV2). The malic enzyme flux (decarboxylating malate dehydrogenases) further shows a marked ~65% increase in S phase (Fig 5H, Appendix Fig S7), as evident by the increased lactate m + 3 labeling in S phase when feeding isotopic glutamine (Fig 4E). Consistently, an increased concentration of lactate m + 3 in the culture media is further observed in S phase (Appendix Fig S11). Notably, while glutamine-derived anaplerotic flux increases in S phase, there is no major change in glucose-derived anaplerotic flux through pyruvate carboxylase in S phase (Fig 5I, Appendix Fig S5). This is evident by the drop in malate and aspartate m + 3 in S phase when feeding isotopic glucose (Fig 4F and Appendix Fig S12) occurring while the concentration of malate and aspartate increases (Fig 4A and B).

While the increased glutamine-derived flux into the TCA cycle in S phase supports an increase in α-ketoglutarate oxidation, we further detect a surprisingly high ~55% increase in the rate of α-ketoglutarate reduction in early S phase (Fig 5B, Appendix Fig S3). This is evident by the marked increase in m + 5 citrate in S phase when feeding isotopic glutamine (Fig 4G). Considering the major drop in glycolytic flux into the TCA cycle in S phase, the relative contribution of reductive IDH to citrate production increases from ~15% in G1 to ~24% in S phase. Cell cycle oscillations in the relative contribution of glucose versus glutamine to citrate biosynthesis are further observed in the labeling of acetyl-CoA (produced in cytosol from citrate via ATP citrate lyase), where the fractional labeling of acetyl-CoA m + 2 from [U-$^{13}$C]-glucose peaks in G1 phase (Fig 4H) and acetyl-CoA m + 2 from [U-$^{13}$C]- glutamine in S phase (Fig 4I). Accordingly, acetylated amino acids show increased m + 2 labeling from glucose in G1 and from glutamine in S phase (Appendix Fig S13).

The complementary oscillations in glucose versus glutamine-derived flux into the TCA cycle result in an overall uniform production rate of reducing equivalents (NADH/FADH$_2$) ~85 ± 5 mM/h throughout the cell cycle (Fig 5E, Materials and Methods): Glucose-derived production of reducing equivalents peaks in G1/S while glutamine-derived production of reducing equivalents peaks in late S phase. While the relative contribution of glutamine to NADH/FADH$_2$ production oscillates (between ~60% in G1/S and ~75% in late S phase), it remains the prime source of reducing power for driving oxidative phosphorylation all throughout the cell cycle, in accordance with previous measurements in non-synchronized cells (Fan *et al*, 2013). Consistent with the total production rate of NADH/FADH$_2$ remaining constant throughout the cell cycle, we find that oxygen consumption rate does not change throughout the cell cycle (Appendix Fig S14). Hence, the complementary oscillations in glucose versus glutamine oxidation in TCA cycle result in a constant rate of reducing equivalent production, sustaining a constant rate of mitochondrial oxidative phosphorylation flux throughout the cell cycle.

### Suppression of glycolytic flux into TCA cycle in S phase is important for cellular progression through the cell cycle

The drop in glycolytic flux into TCA cycle in S phase (Fig 5A) involves a major twofold decrease in flux through pyruvate dehydrogenase (PDH), the prime source for acetyl groups for TCA cycle oxidation. PDH is negatively regulated by pyruvate dehydrogenase

kinase (PDK), and treatment with the PDK inhibitor dichloroacetate (DCA) was previously shown to enhance glycolytic flux into TCA cycle while decreasing reductive glutamine metabolism toward citrate biosynthesis (Fendt *et al*, 2013). Accordingly, treating the synchronized HeLa cells with 4 mM of DCA for 1 h leads to a marked increase in the fractional labeling of citrate m + 2 and acetyl-CoA m + 2 from isotopic glucose, representing a major increase in glycolytic flux into the TCA cycle (Fig 6A and B). Notably, DCA treatment completely eliminates the oscillations in glycolytic flux into the TCA cycle, as evident by a uniform fractional labeling of citrate m + 2 and acetyl-CoA m + 2 when feeding DCA throughout the cell cycle. DCA treatment further leads to a uniform citrate m + 5/m + 4 ratio and fractional labeling of acetyl-CoA m + 2 from isotopic glutamine, eliminating the oscillations in glucose versus glutamine flux toward acetyl-CoA production (Fig 6C and D).

To test whether the suppression of glycolytic flux into TCA cycle in S phase is important for progression of cells through the cell cycle, we measured the cell cycle phase distribution in synchronized HeLa cells after 3-h treatment with DCA. We find that DCA treatment leads to 16% increase in the fraction of cells in S phase (Fig 6E, two-tailed *t*-test *P*-value = 0.006). Treating of non-synchronized HeLa cells as well as colon carcinoma cells (HCT116) with DCA for 24 h further shows a significant increase in the fraction of cells in S phase (Fig 6F, 30% increase in HeLa, two-tailed *t*-test *P*-value < 10$^{-3}$, 17% increase in HCT116, *P*-value < 10$^{-5}$). Notably, the observed increase in the fraction of cells in S phase represents slower progression rate of cells through S phase rather than cell cycle arrest at that phase, as almost all HeLa cells (~97%) complete at least one cell cycle after a 72-h treatment with DCA (Appendix Fig S15). Conversely, treating of cells with a mitochondrial pyruvate carrier inhibitor (UK5099), which slows glycolytic flux into TCA cycle, shows the opposite effect of lowering the fraction of cells in S phase (Appendix Fig S16). Overall, our results indicate that the shift from glucose to glutamine-derived flux into TCA cycle plays an important role in cellular progression through S phase.

## Discussion

We described a temporal-fluxomics approach for analyzing the dynamics of intracellular metabolic flux throughout the cell cycle in proliferating human cells. Inferring cell cycle-dependent changes in flux is technically challenging due to several factors including the perturbative nature of synchronization-induced growth arrest, the gradual loss of population synchrony, and the requirement for accurate measurements of oscillations in metabolite pool sizes that in many cases vary by less than twofold at maximum. Addressing these challenges, we tracked synchronized cells for three complete cell cycles, performed LC-MS-based metabolomics and pulse-chase isotope tracing in the synchronized cells, and employed computational deconvolution techniques to reliably detect oscillations in metabolite concentrations and isotopic labeling dynamics. Inferring transient fluxes within each 1-h interval throughout the cell cycle was complicated by the fact that the labeling of TCA cycle intermediates in the synchronized cells does not reach isotopic steady state within 1-h feeding with the isotopic nutrients. This was addressed by modeling the isotopic labeling kinetics of metabolites within each 1-h interval throughout the cell cycle, for each one time interval

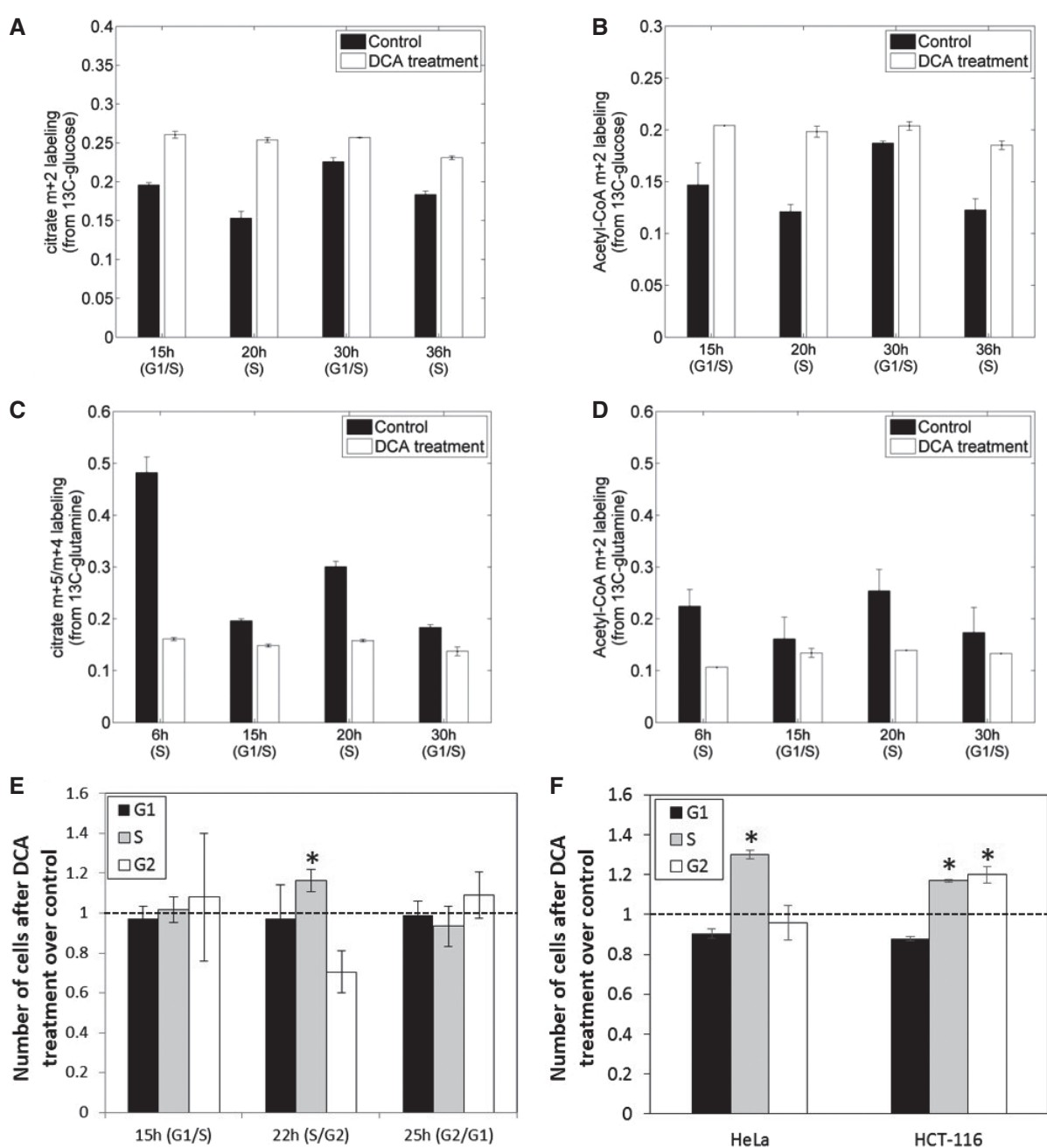

**Figure 6. PDK inhibition via DCA treatment eliminates the oscillation of glycolytic flux into TCA cycle and inhibits cellular progression through S phase.**

A–D    One-hour treatment of synchronized cells with DCA inhibits the oscillations in citrate m + 2 (A) and acetyl-CoA m + 2 (B) from isotopic glucose (representing glycolytic flux into TCA cycle). It further inhibits oscillations in citrate m + 5/m + 4 ratio (C) and acetyl-CoA m + 2 (D) from isotopic glutamine (representing oxidative versus reductive TCA cycle flux).

E    The fraction of cells in G1, S, and G2/M phases in synchronized HeLa cells after 3-h treatment with DCA (normalized by measurements in untreated control cells).

F    The fraction of cells in G1, S, and G2/M phases in non-synchronized HeLa and HCT-116 cells after 24-h treatment with DCA (normalized by measurements in untreated control cells).

Data information: As shown, DCA significantly increases the fraction of cells in S phase, inhibiting cellular progression into G2 phase, showing mean and s.d. of $n = 3$ for all isotopic labeling forms and FACS measurements (A–F). Statistical significance of changes in the fraction of cells in each cell cycle phase following DCA treatment is calculated based on two-tailed, unequal variance *t*-test (E, F; *P-value < 0.01).

using an approach conceptually similar to non-stationary metabolic flux analysis (MFA; Noh *et al*, 2006; Noack *et al*, 2011). Applied to HeLa cells, we derived a first comprehensive and quantitative view

of metabolic flux oscillations at a high temporal resolution in central metabolism throughout the cell cycle of human cells, showing complementary oscillations between glucose and glutamine-derived

flux in the TCA cycle throughout the cell cycle. Cell cycle-dependent changes in flux through the pentose-phosphate pathway were previously studied using isotope tracing in synchronized human cell lines (Vizan *et al*, 2009) and in yeast by also utilizing MFA (Costenoble *et al*, 2007).

The inferred flux oscillations through central metabolism via glucose and glutamine could potentially be biased by oscillations in the metabolism of other carbon sources. For example, the uptake and catabolism of glucogenic and ketogenic amino acids may feed into TCA cycle and potentially oscillate through the cell cycle. Though, apparently, the relative contribution of amino acid catabolism to TCA cycle flux in HeLa cells is extremely low; feeding isotopic glucose and glutamine for 24 h shows that more than 97% of the carbons in TCA cycle metabolites are derived exclusively from glucose and glutamine (and from atmospheric $CO_2$, Appendix Fig S17). Hence, a potential change in amino acid metabolism throughout the cell cycle would have little or no effect on the reported flux oscillations from glucose and glutamine. Not accounting for potential oscillations through other reactions implicated in central metabolism of glucose and glutamine could in principle also bias the presented flux estimations. Though, notably, the flux analysis here is performed separately through groups of converging reactions producing different metabolites (e.g., for reactions producing citrate, malate/oxaloacetate, and lactate); hence, a potential bias in some of these independent flux estimations would not change the overall emerging view of glucose and glutamine oscillations; for example, the increased TCA cycle metabolism of glutamine in S phase is supported by several independent flux estimations showing increased glutamine-derived flux into TCA cycle in S phase, oxidation of α-ketoglutarate, reduction of α-ketoglutarate, malic enzyme flux, and increased nucleotide biosynthesis. Another simplifying assumption that facilitated the estimation of flux dynamics throughout the cell cycle is of rapid mixing of mitochondrial and cytosolic metabolite pools, giving rise to estimates of whole-cell level fluxes. This assumption is typically made when analyzing flux in eukaryotic cells due to experimental complications in measuring metabolite concentrations and labeling dynamics in distinct subcellular compartments. Methodological advancements in subcellular level metabolomics are required for further studies on cell cycle oscillations in metabolic flux in mitochondria versus cytosol.

Our result of an induced glycolytic flux in G1/S phase is qualitatively consistent with previous reports ~3- to 10-fold increase in glycolytic flux in G1/S in HeLa cells (Colombo *et al*, 2011; Tudzarova *et al*, 2011). However, here, analyzing the metabolism of synchronized cells after resuming exponential growth (i.e., completing an entire cell cycle after released from synchronization growth arrest) suggests a moderate increase in glycolytic flux of only ~65% in G1/S phase. The smaller magnitude of the oscillations in glycolytic flux inferred here is in agreement with the moderate changes in concentration of glycolytic intermediates, which increase only ~2-fold in G1/S phase. Our finding of increased glutamine-derived flux into the TCA cycle and support of pyrimidine biosynthesis in S phase agree with reports of the essentiality of glutamine (and not glucose) for entering and progressing through S phase, which can be rescued by nucleotide feeding (Gaglio *et al*, 2009; Colombo *et al*, 2011).

We showed that complementary oscillations in the rate of glucose versus glutamine oxidation in the TCA cycle result in a constant rate of $NADH/FADH_2$ production and reduction of oxygen by the electron transport chain throughout the cell cycle. Notably, while here we observe a constant rate of oxygen consumption throughout the cell cycle, fluctuations in oxygen consumption were previously reported in yeast cells, with DNA replication and cell division occurring when oxygen consumption rate is low, protecting genome integrity (Chen *et al*, 2007). On the other hand, respiration was suggested to actually protect against oxygen-associated DNA damage in proliferating human cells by reducing the intracellular oxygen concentration and ROS levels (Sung *et al*, 2010) and hence may potentially be beneficial during S phase. Cell cycle-dependent changes in the utilization of glucose versus glutamine may be associated with reported changes in mitochondrial structure throughout the cell cycle, converted from isolated fragments into a hyperfused network in G1/S transition (Mitra *et al*, 2009).

While the current study focuses on identifying and quantifying oscillations in metabolic flux throughout the cell cycle, further research is required to decipher how these metabolic changes are regulated. The ubiquitin ligase complexes APC/C and SCF complex were claimed to control glycolytic flux by limiting the expression of PFKFB3 to the G1/S transition, in accordance with the identified increase in glycolytic flux. SCF complex further limits the expression of GLS1 to S and G2/M phases, in agreement with our finding of induced glutamine to glutamate conversion in these cell cycle phases. More generally, metabolic enzymes are typically not regulated at the level of mRNA or protein throughout the cell cycle based on cell cycle transcriptomics and proteomics studies (Olsen *et al*, 2010). However, post-translational modification of central metabolic enzymes is highly abundant and oscillations in phosphorylation levels of metabolic enzymes have been described (Olsen *et al*, 2010). Considering our finding of oscillations in the concentration of numerous metabolic intermediates, as well as energy and redox cofactors, suggests that metabolic regulation via changes in enzyme-binding site occupancy and allosteric regulation may also play a key role in regulating cell cycle flux dynamics. This is further supported by the fact that metabolic substrate and inhibitor levels in mammalian cells are typically in the same range as the $K_m$ values for the corresponding enzymes (Park *et al*, 2016).

We showed that treatment of HeLa cells with the PDK inhibitor DCA eliminates the oscillations in glucose flux into TCA cycle, suggesting that cell cycle-specific regulation of PDH activity may be involved in regulating these flux oscillations. Consistently, PDK4 was reported to be induced by the E2F-pRB pathway which controls cell entry to S phase (Hsieh *et al*, 2008; Kaplon *et al*, 2015). Furthermore, analyzing published phosphoproteomics data for HeLa cells measured throughout the cell cycle (Olsen *et al*, 2010) shows more than twofold increase in the phosphorylation of the PDH E1 component in early and late S phase versus in G1 (Appendix Fig S18). Validating this observation here, we find a significant twofold increase in the phosphorylation of PDH- S232 specifically in S phase (Appendix Fig S19). A drop in glycolytic flux into TCA cycle in S phase is further supported by a recent report of a drop in the abundance of the pyruvate dehydrogenase complex in mitochondria in S phase, following translocation of the complex components to the nucleus (Sutendra *et al*, 2014). Further research is required to determine the precise regulatory

mechanism that underlies cell cycle oscillations in glycolytic flux into TCA cycle, which may potentially spread quantitatively among several enzymes [in accordance with the view of metabolic control analysis (Fell, 1997)].

In the conventional view of mammalian metabolism, acetyl-CoA (a major anabolic precursor for fatty acid biosynthesis) is primarily produced by the oxidation of glucose-derived pyruvate in mitochondria. Previous studies have employed isotope tracers to show that in cancer cells grown under hypoxia (Metallo *et al*, 2012), in cells with defective mitochondria (Mullen *et al*, 2012), and in anchorage-independent growth (Jiang *et al*, 2016), a major fraction of acetyl-CoA is produced via another route, reductive carboxylation of glutamine-derived α-ketoglutarate (catalyzed by reverse flux through isocitrate dehydrogenase, IDH). Under these conditions, feeding cells with isotopic glutamine leads to a marked increase in the fractional labeling of citrate m + 5 and consequently in the isotopic labeling of synthesized fatty acids. Here, we showed that the fractional labeling of citrate m + 5 significantly oscillates throughout the cell cycle under standard normoxic conditions. This reflects major oscillations in the relative contribution of oxidative TCA cycle flux (peaking in G1) and in the reductive metabolism of glutamine-derived α-ketoglutarate (peaking in S) to the production of acetyl-CoA throughout the cell cycle. Notably, though, the oxidative IDH flux remains several-fold higher than the reductive flux all throughout the cell cycle, reflecting an overall net flux in the oxidative direction.

Understanding the metabolic adaptation of cells to tumorigenic mutations is a central goal of cancer metabolic research. Considering that tumorigenic mutations typically alter cell cycle progression, flux alterations observed at a cell population level may merely reflect a change in the distribution of cell cycle phases in the population (due to cells in different phases having different metabolic fluxes). Hence, the presented temporal-fluxomics approach will enable to revisit our understanding of oncogene-induced metabolic alterations, disentangling population level artifacts from directly regulated flux alterations with important tumorigenic role and revealing potential targets for therapy. Combined targeting of cell cycle-specific flux alterations with drugs that block progression through the same cell cycle phases is expected to have important therapeutic applications (Diaz-Moralli *et al*, 2013; Saqcena *et al*, 2015).

# Materials and Methods

## Materials

Most of the chemical reagents were purchased from Sigma-Aldrich unless otherwise specified. Stable isotopes [U-$^{13}$C]-glucose and [U-$^{13}$C]-glutamine were obtained from Cambridge Isotope Laboratories, Inc. Cell culture media and reagents were purchased from Biological Industries, unless otherwise specified. HeLa and HCT-116 cells were purchased from ATCC. For oxygen consumption measurements, chemicals (inhibitors and culture medium) were obtained from Sigma and other materials from Agilent. Antibodies used were as follows: phospho S232 PDH E1 alpha protein (PDHA1) Profiling ELISA Kit, (ABcam-ab115343); human PDH E1 alpha protein ELISA kit (PDHA1), (Abcam-ab181415).

## Cell culture and synchronization

HeLa cells were cultured in Dulbecco's modified Eagle's medium (high glucose, Biological Industries, 01-055-1A) supplemented with 10% (v/v) heat-inactivated dialyzed fetal bovine serum (Biological Industries), 3 mM L-glutamine, 100 U/ml penicillin, and 100 μg/ml streptomycin with 5% CO$_2$ in a humidified incubator at 37°C. Culture medium was additionally supplemented with 84 mg/ml L-serine, and 48 mg/ml L-cystine to maintain sufficient amount of nutrients for three cell doublings. HeLa cells have been validated by the vendors, and we tested for mycoplasma using EZ-PCR mycoplasma detection kit (Biological Industries). Cell number and volume analysis were performed using a Z2 Beckman Coulter Counter (100 μm aperture); cells were trypsinized and resuspended in IsoFlow Sheath Fluid (Beckman Coulter) immediately before counting.

Cell synchronization was achieved using double thymidine block. Briefly, 2 mM thymidine (Sigma-T1895) was added to 10-cm culture plates at 25–30% confluence for 17 h. Cells were released from the first block by washing twice with phosphate buffer saline (PBS) and replacing with fresh culture medium. After 9 h, cells were incubated with 2 mM thymidine for a second block time for 17 h. Cells were replated in smaller plates (60 or 35 mm) for further analysis. Cell cycle analysis of synchronized cells was performed by quantitation of DNA content using propodium iodide (PI) staining followed by flow cytometry. For PI staining, cells were fixed using 75% ethanol/PBS and then resuspended in 0.5 ml of PI staining solution (3.8 mM sodium citrate, 40 μg/ml PI, 50 ng/ml RNase A) for 40 min at room temperature in dark. Flow cytometric analysis was performed using LSRII (BD Biosciences, with at least 50,000 cells per FACS run). Cell cycle stages from raw FACS data were quantified using Modfit (Verity House Software).

To check the effect of dichloroacetate treatment on cell cycle progression, synchronized cells were treated with 4 mM pyruvate dehydrogenase kinase inhibitor dichloroacetate (Sigma-Aldrich) for 3 h. Non-synchronized cells were treated with 16 mM DCA for 24 h, followed by cytometric analysis after DNA staining with PI.

## LC-MS-based metabolomics and isotope tracing

To measure intracellular metabolite pools, cells were washed with 2 ml of ice-cold PBS for three times and metabolites extracted with 200 μl of 50:30:20 (v/v/v) methanol:acetonitrile:water solution at −20°C. The cells were quickly scraped on dry ice. For the extraction of metabolites from the culture medium, 50 μl of media was mixed with 200 μl of 50:30 (v/v) methanol:acetonitrile solution at −20°C. All metabolite extractions were stored at −80°C for at least 1 h, followed by centrifugation, twice at 20,000 *g* for 20 min to obtain protein-free metabolite extraction.

Metabolite pool sizes are expressed per total cell volume, measured using a Coulter counter in cells grown in parallel in different plates. Absolute metabolite concentrations for specific metabolites of interest were determined based on isotope ratio using chemical standards (Bennett *et al*, 2008; Dataset EV3). Pulse isotopic labeling was performed by feeding synchronized cells at each time point with either [U-$^{13}$C]-glucose or [U-$^{13}$C]-glutamine for 1 h. To minimize the perturbation to cells due to the replacement with fresh media, we used conditioned medium obtained from a

previous culture of HeLa cells. Specifically, conditioned medium with either isotopic glucose or isotopic glutamine was incubated with fully attached HeLa cells at ~30% confluence for 4 h and then stored in 4°C until used for pulse-chase labeling experiments.

Chromatographic separation was achieved on a SeQuant ZIC-pHILIC column (2.1 × 150 mm, 5 μm bead size, Merck Millipore). Flow rate was set to 0.2 ml/min, column compartment was set to 30°C, and autosampler tray was maintained at 4°C. Mobile phase A consisted of 20 mM ammonium carbonate with 0.01% (v/v) ammonium hydroxide. Mobile Phase B was 100% acetonitrile. The mobile phase linear gradient (%B) was as follows: 0 min 80%, 15 min 20%, 15.1 min 80%, and 23 min 80%. A mobile phase was introduced to Thermo Q-Exactive mass spectrometer with an electrospray ionization source working in polarity switching mode. Metabolites were analyzed using full-scan method in the range 70–1,000 m/z and with a resolution of 70,000. Positions of metabolites in the chromatogram were identified by corresponding pure chemical standards. Data were analyzed with MAVEN (Clasquin *et al*, 2012).

## Measurement of oscillations in oxygen consumption

Measurement of oxygen consumption was done using the XFp Extracellular Flux Analyzer (Agilent). Briefly, HeLa cells after synchronization were plated (20,000 cells/well) into XFp culture mini plates and grown at 37°C with 5% $CO_2$ in a humidified incubator for various times to enrich the cell population with cells at distinct cell cycle phases. Cells were washed and incubated with prewarmed XF assay medium (Sigma D5030, pH 7.4) supplemented with 25 mM glucose and 3 mM glutamine for 1 h in a non-$CO_2$ incubator at 37°C. Appropriate dilutions of the inhibitors (final well concentrations: oligomycin 1 μM, FCCP 1 μM, rotenone/antimycin A 1 μM) were prepared in the assay medium as per the instructions in the manual. Hydrated sensor cartridges were calibrated prior to the measurement on SeaHorse XFp Extracellular Flux Analyzer (Agilent). Data acquisition consisted of a baseline measurement followed by oligomycin, FCCP, and rotenone/antimycin A injections, respectively. OCR data were normalized against cell volume obtained from Coulter counter measurements of the cells from a parallel plate without any treatment and expressed in pmoles/min/μl.

## Synchronization loss model

We construct a probabilistic model that describes the loss of synchronization following release from double thymidine block, due to cell–cell variability in the rate of progression through the cell cycle. Each cell is assumed to have its own "internal clock", which controls the speed at which it progresses through the cell cycle (denoted by $\gamma$). The relative progression rates of cells released from synchronization arrest are assumed to be normally distributed: $\gamma \sim N(1, \sigma^2)$. We estimate the variance of the distribution ($\sigma^2$) as well as the duration of G1, S, and G2/M (denoted by $d_G$, $d_S$, and $d_M$, respectively, in hours), given the FACS measurements of the fraction of cells in each cell cycle phase in the synchronized cell population (as described below). We denote the cell doubling time by $d_{CYC}$ ($= d_G + d_S + d_M$, in hours). For a cell whose rate of progression through the cell cycle is $\gamma$, the cell-intrinsic time $x$ (in hours) within the cell cycle ($0 \leq x \leq d_{CYC}$) at time $t$ after the release from synchronization-induced growth arrest is

$$x = \gamma \cdot t + d_G - \left\lfloor \frac{(\gamma \cdot t + d_G)}{d_{CYC}} \right\rfloor d_{CYC}, \tag{1}$$

considering that cells resume growth in G1/S transition after released from double thymidine block. For example, for a cell with a relative progression rate through the cell cycle of $\gamma = 1$ released from growth arrest, it will take $d_S + d_M$ hours to complete one cell cycle (and then have an intrinsic time of $x = 0$), while for a cell with double the rate ($\gamma = 2$), it will take half the time (i.e., $(d_S + d_M)/2$). At time $t$ after the release from synchronization arrest, a cell having an intrinsic time of $x$ has a relative progression rate through the cell cycle $\gamma$ equal to $\frac{1}{t}(x + k \cdot d_{CYC} - d_G)$, with $k$ representing the number of completed cell cycles since the release from growth arrest (based on equation 1). Hence, considering that $\gamma$ is normally distributed, we can compute the number of cells in the synchronized population at time $t$ whose cell-intrinsic time is $x$ (denoted $g'(x,t)$) as:

$$g'(x,t) = \sum_{k=0}^{2} 2^k \cdot e^{-\frac{\left(\frac{x + k \cdot d_{CYC} - d_G}{t} - 1\right)^2}{2\sigma^2}}, \tag{2}$$

considering values of $k$ between zero and two, representing three complete cell cycle. We denote by $g(x,t)$ the probability density function of the number of cells at time $t$ whose cell-intrinsic time is $x$, with $g(x,t) = \frac{1}{c} g'(x,t)$, where $C$ is a normalization factor. The expected fraction of the cells in S, G1, and G2/M phases at time $t$ (denoted by $m_s(t)$, $m_G(t)$ and $m_M(t)$, respectively) are calculated as:

$$m_s(t) = \int_0^{d_s} g(x,t)\, dx, \tag{3}$$

$$m_G(t) = \int_{d_s}^{d_s + d_G} g(x,t)\, dx, \tag{4}$$

$$m_M(t) = \int_{d_s + d_G}^{d_{CYC}} g(x,t)\, dx, \tag{5}$$

We perform a maximum log-likelihood estimation of the four parameters of the model ($\sigma$, $d_G$, $d_S$, and $d_M$), minimizing the variance-weighted sum of squared residuals between the simulated fraction of cells in the different cell cycle phases ($m_s(t)$, $m_G(t)$, and $m_M(t)$) and the FACS measurements (assuming Gaussian noise in FACS measurements with an empirically estimated standard deviation of ~10%). This minimization was performed via an implementation of sequential quadratic optimization (SQP) available in MATLAB. Confidence intervals were computed by the likelihood ratio test, comparing the maximum log-likelihood estimates with that obtained when constraining each of the four parameters to increasing and then decreasing value (considering the 95% quantile of chi-squared distribution with one degree of freedom). The optimal parameters found were $\sigma = 11\% \pm 1\%$, $d_G = 6.8$ h $\pm 0.8$ h, $d_S = 6.4$ h $\pm 0.5$ h, and $d_M = 3.2$ h $\pm 0.4$ h. Overall, the good fit between the model prediction and experimental data shown in Fig 1B supports the underlying assumptions of this model. Computing the duration of each cell cycle phase based on PI staining/FACS measurements in a population of non-synchronized HeLa cells and considering a decreasing exponential cell age distribution show 8.2 h for G1, 4.9 h for S, and 2.9 h for G2/M. The small under

estimation of the duration of G1 and over estimation of the duration of S by the analysis of the synchronized cell (both not more than 1 h off the measurements in the non-synchronized cells) may be due to a slight perturbation to cell cycle dynamics due to synchronization-induced growth arrest.

## Computational deconvolution of cell volume measurements in the synchronized cell population

Computational deconvolution was used to estimate cell volume dynamics throughout the cell cycle, correcting for cell dispersion that bias Coulter counter measurements of cell volume performed in the synchronized cells. Specifically, denoting the average cell volume in the synchronized cell population at time $t$ by $v(t)$ ($9 \leq t \leq 45$), we estimate the average cell volume in the cell-intrinsic time $x$ within the cell cycle ($0 \leq x \leq d_{CYC}$), denoted by $v'(x)$, as following:

$$v(t) = \int_0^{d_{CYC}} v'(x)g(x,t)\,\mathrm{d}x + \varepsilon(t), \tag{6}$$

where $\varepsilon(t)$ represents experimental noise in the volume measurement performed at time $t$. We represent $v'(x)$ using a cubic spline, which is a commonly used approach for fitting biological time-series data (considering splines with four segments, defined based on five knots). Non-convex optimization was used to find the optimal position of the knots and corresponding value of $v'(t)$ minimizing the sum-of-square of the error terms. Non-convex optimizations were solved using MATLAB's implementation of SQP.

## Computational deconvolution of metabolite concentration, isotope labeling, and uptake and secretion rate measurements

Given a metabolite $i$ whose measured concentration in the synchronized cell population at time $t$ ($9 \leq t \leq 45$) is denoted by $u_i(t)$ (measured metabolite pool size normalized by the measured average cell volume at time $t$, $v(t)$), we estimate the concentration in cell-intrinsic time $x$ within the cell cycle ($0 \leq x \leq d_{CYC}$), denoted by $u'_i(x)$ as:

$$u_i(t) = \frac{1}{\int_0^{d_{CYC}} v'(x)g(x,t)\,\mathrm{d}x} \int_0^{d_{CYC}} u'_i(x)v'(x)g(x,t)\,\mathrm{d}x + \varepsilon(t), \tag{7}$$

considering that the measured concentration of metabolite $i$ at time $t$ represents the average concentration in cells with cell-intrinsic time $x$, weighted by the total volume of cells with intrinsic time $x$ at time $t$ (i.e., $v'(x)g(x,t)$). We represent $u'_i(x)$ using a cubic spline and estimate its coefficients as described above.

We denote the measured relative abundance of the $k^{\text{th}}$ mass isotopomer of metabolite $i$ (i.e., the fraction of the metabolite pool having $k$ labeled carbons) after 1-h feeding with an isotopic substrate of synchronized cells at time $t$ by $u_{i,k}(t)$. We estimate the relative abundance of the $k^{\text{th}}$ mass isotopomer of metabolite $i$ in cell-intrinsic time $x$ within the cell cycle denoted $u'_{i,k}(x)$ as:

$$u_{i,k}(t) = \frac{1}{\int_0^\infty u'_i(x)v'(x)g(x,t)\,\mathrm{d}x} \int_0^\infty u'_{i,k}(x)u'_i(x)v'(x)g(x,t)\,\mathrm{d}x + \varepsilon(t), \tag{8}$$

considering that the measured fractional isotopic labeling of a metabolite $i$ at time $t$ represents the average labeling in cells with intrinsic cell cycle time $x$, weighted by the metabolite pool size in cells with cell-intrinsic time $x$ (i.e., with $u'_i(x)v'(x)g(x,t)$).

We denote the measured change in pool size of metabolite $i$ in the culture media between time $t-\Delta t$ and time $t$ by $\Delta e_i(t)$. We estimate the transport flux of metabolite $i$ by the synchronized cell population at time $t$, denoted $f_i(t)$ (in molar amount per unit of cell volume per hour, with positive and negative flux representing secretion and uptake, respectively) by dividing the change in pool size of metabolite $i$ by the accumulated volume of cells in the culture metabolite between time $t-\Delta t$ and time $t$:

$$f_i(t) = \frac{1}{\int_{t-\Delta t}^{t} \int_0^{d_{CYC}} v'(x)g(x,t')\,\mathrm{d}x\mathrm{d}t'}\Delta e_i(t), \tag{9}$$

The transport flux of metabolite $i$ at cell-intrinsic time $x$, denoted $f'_i(x)$ is estimated as:

$$\Delta e_i(t) = \int_{t-\Delta t}^{t} \int_0^{d_{CYC}} f'_i(x)v'(x)g(x,t')\,\mathrm{d}x\mathrm{d}t' + \varepsilon(t), \tag{10}$$

considering that the measured change in pool size of metabolite $i$ at time $t$ represents the cumulative transport within the $\Delta t$ time interval by cells with different intrinsic cell cycle time $x$.

## Statistical significance of oscillations in metabolite concentrations and isotopic labeling patterns

To assess the statistical significance of observed oscillations in the deconvoluted concentration of a certain metabolite, we compared the observed amplitude of the oscillation to the amplitude expected by chance (considering the noise in LC-MS measurements). Specifically, for each metabolite $i$, we define the amplitude of its oscillation by $a_i$ as:

$$a_i = \max_{0 \leq x \leq d_{CYC}} u'_i(x) - \min_{0 \leq x \leq d_{CYC}} u'_i(x). \tag{11}$$

Next, we compute the distribution of amplitudes expected by chance by repeating the following steps 10,000 times: For each time $t$ for which LC-MS measurements were performed on the synchronized cell population ($9 \leq t \leq 45$), we generate a random metabolite concentration [denoted $r(t)$] by sampling from a normal distribution whose mean is the average concentration of metabolite $i$ measured throughout all time points in the synchronized cells, and with the standard deviation of the experimental measurement of metabolite $i$ at time $t$, denoted $\sigma_i(t)$:

$$r(t) \sim N\left(\frac{1}{k}\sum_{t=1}^{k} u_i(t), \sigma_i^2(t)\right). \tag{12}$$

Computational deconvolution (equation 7) is applied on the randomly generated metabolite concentration data [i.e., $r(t)$] and an empirical *P*-value computed based on the fraction of randomly sampled concentration vectors for which the derived amplitude is equal or larger than that computed for the concentration measurements of metabolite $i$. FDR correction for multiple testing is computed using the approach of Benjamini–Hochberg. A

conceptually similar method was employed to assess the statistical significance of oscillations in the relative abundance of metabolite isotopic labeling [applying computational deconvolution (equation 8) to randomly generated isotopic labeling data].

## Computational inference of metabolic flux dynamics throughout the cell cycle

For every 1-h interval $j$ throughout the cell cycle $j \in \{0 \ldots \lfloor d_{CYC} \rfloor\}$ (referred to as the $j^{th}$ *cell cycle interval*), we computed the most likely momentary fluxes through nine reactions in TCA cycle and in branching pathways (Fig 3B). Toward this end, we employed a variant of kinetic flux profiling (KFP) to separately infer fluxes producing each metabolite $i$ in cell cycle interval $j$, for which the simulated isotopic labeling kinetics optimally match the experimental measurements.

The relative abundance of the $k^{th}$ mass isotopomer of metabolite $i$ after 1-h feeding of cells within cell cycle interval $j$ was inferred based on the pulse-chase labeling experiments in the synchronized cells followed by deconvolution as described above (denoted by $u'_{i,k}(j)$). To estimate the dynamics of the isotopic labeling form of this metabolite within the 1-h cell cycle interval (i.e., within time periods shorter than 1 h), we performed pulse-chase labeling experiments with isotopic glucose and glutamine in non-synchronized cells, measuring the relative abundance of the $k^{th}$ mass isotopomer of metabolite $i$ at different times $t \in T = \{10,20,30,60\}$ (in minutes), denoted by $X_{i,k}(t)$. These measurements were used to estimate the relative abundance of the $k^{th}$ mass isotopomer of metabolite $i$, $t$ minutes after the beginning of the $j^{th}$ 1-h time interval within the cell cycle (denoted by $X_{i,k}^{j}(t)$) by scaling the measurements performed on the non-synchronized cells:

$$X_{i,k}^{j}(t) = \frac{u'_{i,k}(j+1)}{X_{i,k}(60)} X_{i,k}(t). \tag{13}$$

To validate these estimated labeling kinetics, we performed rapid pulse-chase labeling experiments (10 and 30 min) in synchronized cells grown for 15 h (G1/S) and 20 h (S), finding a good match between the estimated and measured labeling dynamics (Appendix Fig S20).

We describe the inference of metabolic flux through reactions producing citrate while other fluxes are obtained similarly (see below). The analysis accounts for citrate synthase ($v_1$) and reductive isocitrate dehydrogenase (IDH, $v_2$) producing citrate (Fig 3B). We denote the total citrate consumption flux by $v_{out}$, which may be lower or higher than the sum of $v1$ and $v2$ in case citrate is accumulated or depleted within a cell cycle interval, respectively (as the synchronized cells are not in metabolic steady state). The expected mass-isotopomer distribution of citrate after $t$ minutes into the $j^{th}$ cell cycle interval, considering the fluxes $v_1$, $v_2$, and $v_{out}$, is denoted $Y_{cit}^{j}(t, v_1, v_2, v_{out})$. Assuming that the error in the measured isotope labeling data is normally distributed, maximum likelihood estimate of fluxes is obtained by minimizing the variance-weighted sum of squared residuals between measured and computed mass-isotopomer distributions, where $\sigma_{cit,k}^{j}$ is the standard deviation in the measurement of the relative abundance of the $k^{th}$ mass isotopomer of citrate in the $j^{th}$ time interval:

$$\min_{v_1, v_2, v_{out}} \sum_{t \in T} \sum_{k \in \{4,5\}} \left( \frac{X_{cit,k}^{j}(t) - Y_{cit,k}^{j}(t, v_1, v_2, v_{out})}{\sigma_{cit,k}^{j}} \right)^2 \tag{14}$$

s.t.

$$v_{out} = v_1 + v_2 + \left( u'_{cit}(j+1) - u'_{cit}(j) \right), \tag{14.1}$$

$$v_1, v_2, v_{out} \geq 0, \tag{14.2}$$

where $u'_{cit}(j)$ represents the deconvoluted concentration of citrate (in mM) in the $j^{th}$ cell cycle interval and is used to constrain the difference between the total citrate producing and consuming flux within the $j^{th}$ 1-h cell cycle interval (equation 14.1). We accounted for the two major mass isotopomers of citrate, m + 4 and m + 5 (Appendix Fig S3). To simulate the labeling kinetics of the $k^{th}$ mass isotopomer of citrate within the $j^{th}$ cell cycle interval, denoted by $Y_{cit,k}^{j}(t, v_1, v_2, v_{out})$, we utilized the following system of ordinary differential equations:

$$\frac{dY_{cit,k}^{j}(t, v_1, v_2, v_{out})}{dt}$$
$$= \frac{1}{u'_{cit}(j)} \left( v_1 X_{mal,k}^{j}(t) + v_2 X_{aKG,k}^{j}(t) - v_{out} Y_{cit,k}^{j}(t, v_1, v_2, v_{out}) \right),$$

where $X_{mal,k}^{j}(t)$ and $X_{aKG,k}^{j}(t)$ represent the relative abundance of the $k^{th}$ mass isotopomer of malate and α-ketoglutarate, $t$ minutes into the $j^{th}$ cell cycle interval, and $v_1 X_{mal,k}^{j}(t)$ and $v_2 X_{aKG,k}^{j}(t)$ represent the momentary production of the $k^{th}$ mass isotopomer of citrate from of malate and from α-ketoglutarate. The term $v_{out} Y_{cit,k}^{j}(t, v_1, v_2, v_{out})$ represents the total momentary consumption of the $k^{th}$ mass isotopomer of citrate. The difference between the momentary production and consumption rate of the different mass isotopomers of citrate (term in parenthesis on right hand side of the equation) is normalized by the concentration of citrate to give the momentary change in fractional labeling.

A similar approach was employed to infer fluxes through reactions producing the following metabolites: (i) *α-ketoglutarate*—rapid isotopic exchange with glutamate results in essentially a single intracellular pool of α-ketoglutarate and glutamate (as reflected by similar labeling kinetics of the two metabolites). We considered α-ketoglutarate/glutamate m + 5 production by oxidative IDH from citrate m + 5 when feeding isotopic glutamine (reaction $v3$ in Fig 3B) and from glutamine m + 5 when feeding isotopic glutamine ($v8$), as well α-ketoglutarate/glutamate m + 3 production by oxidative IDH from citrate m + 4 when feeding isotopic glutamine ($v3$) (Appendix Fig S4), (ii) *Malate*—considering the rapid isotopic exchange with aspartate (> 100 mM/h based KFP analysis of malate and aspartate labeling kinetics, we account for a single malate/aspartate pool. We consider for malate/aspartate m + 4 production from α-ketoglutarate m + 5 ($v4$), when feeding isotopic glutamine, and malate/aspartate m + 4 production from pyruvate m + 3 ($v7$), when feeding isotopic glucose (Appendix Fig S5), (iii) *UTP*—considering UTP m + 3 production from carbamoyl-aspartate m + 3 when feeding isotopic glutamine ($v5$) (Appendix Fig S6), (iv) *Lactate*—considering lactate m + 3 production by malic enzyme when feeding isotopic glutamine ($v6$) and the production of non-labeled lactate by glycolysis (Appendix Fig S7). We consider an average

malic enzyme flux of 7.9 mM/h throughout the cell cycle (considering a fractional labeling of 1.3% m + 3 lactate under isotopic steady state, when feeding isotopic glutamine, and lactate secretion rate of 610 mM/h).

Non-convex optimizations were solved using MATLAB's implementation of sequential quadratic optimization (SQP). All optimizations were run 10 times, starting from different sets of random fluxes, to overcome potential local minima. To compute confidence intervals for estimated fluxes, we iteratively ran the SQP optimization to compute the maximum log-likelihood estimation while constraining the flux to increasing (and then decreasing) values (with a step size equal to 5% of the flux predicted in the initial maximum log-likelihood estimation; Antoniewicz *et al*, 2006; Fan *et al*, 2013). The confidence interval bounds were determined based on the 95% quantile of chi-squared distribution with one degree of freedom. Notably, while all flux estimates are given in mM/h (i.e., fmole/pl-cells/h), multiplying a flux estimate with intrinsic time $x$ with the estimated cell volume at that time (i.e., $v'(x)$ in pl, see Fig 1C) gives a flux value per cell (fmole/cell/h).

The rate of production of reducing equivalents for driving oxidative phosphorylation generated by glucose oxidation was calculated by summing the flux through the following NADH producing reactions: PDH (according to reaction $v1$ in Fig 3B, considering that ~99% of pyruvate is produced by glucose oxidation throughout the cell cycle, with the fractional labeling of pyruvate from isotopic glutamine under isotopic steady state being < 0.01), oxidative IDH (reaction $v3$), and the rate of shuttling of NADH produced in glycolysis for oxidation in mitochondria (estimated based on pyruvate secretion, a uniform flux of ~17 mM/h measured throughout the cell cycle). The rate of reducing equivalents production from glutamine oxidation was calculated based on the rate of α-ketoglutarate oxidation in TCA cycle (reaction $v4$ in Fig 3B, considering that > 95% of α-ketoglutarate is produced from glutamine all throughout the cell cycle; the fractional labeling of α-ketoglutarate from isotopic glutamine under isotopic steady state is > 0.95), producing NADH by α-ketoglutarate dehydrogenase and $FADH_2$ by succinate dehydrogenase (SDH). Malate dehydrogenase ($v4$ in Fig 3B) further produces another NADH.

### Data availability

- Oscillations in the concentrations of 57 metabolites throughout the cell cycle: Dataset EV1.
- Oscillations in the relative abundance of metabolite mass isotopomers when performing pulse-chase isotope tracing experiments with [U-$^{13}$C]-glucose and [U-$^{13}$C]-glutamine in synchronized HeLa cells: Dataset EV2.
- Concentrations of metabolites in non-synchronized, exponentially growing HeLa cells (in mM): Dataset EV3.

**Expanded View** for this article is available online.

### Acknowledgements

We would like to thank Jing Fan and Won Dong Lee for providing valuable comments on this manuscript. The research leading to these results has received funding from the European Research Council/ERC Grant Agreement No. 714738, and the Israel Science Foundation (ISF), Grant No. 1717/16. AT is funded by the Israel Cancer Research Fund (ICRF), Grant No. RCDA00102, the Israeli Centers of Research Excellence (I-CORE), Gene Regulation in Complex Human Disease, Center no. 41/11, and the Israel Science Foundation (ISF), Grant No. 659/16.

### Author contributions

EA and PK performed the biological experiments. DM assisted with LC-MS work. EA, PK, and TS designed the research, analyzed data, and wrote the article. AT participated in the research design.

### Conflict of interest
The authors declare that they have no conflict of interest.

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
