## [Review Process File · Molecular Systems Biology]

Temporal-fluxomics reveals oscillations in TCA cycle flux throughout the mammalian cell cycle

Eunyong Ahn, Praveen Kumar, Dzmitry Mukha, Amit Tzur & Tomer Shlomi

Corresponding author: Tomer Shlomi, Technion

Review timeline:

Submission date:	24 May 2017
Editorial Decision:	30 June 2017
Revision received:	22 August 2017
Accepted:	25 September 2017

Editor: Maria Polychronidou

Transaction Report:

1st Editorial Decision

30 June 2017

Thank you again for submitting your work to Molecular Systems Biology. We have now heard back from the three referees who agreed to evaluate your study. As you will see below, the reviewers think that the study is interesting. They raise however a series of concerns, which we would ask you to address in a revision of the manuscript.

I think that the reviewers' recommendations are rather clear. All three of them provide constructive suggestions on how to improve the manuscript and therefore I think that there is no need to repeat the points listed below. Of course, feel free to contact me in case you would like to discuss any particular point in further detail.

REVIEWER REPORTS

Reviewer #1:

Ahn and colleagues perform a cell cycle resolved flux analysis in HELA cells synchronized with a double thymidine block. They find that glucose oxidation, lactate secretion and glutamine metabolism change dependent on the cell cycle phase. Moreover, they find that at least part of the change in glucose metabolism during the cell cycle is PDK dependent. This work is interesting and relevant despite being limited in providing a mechanistic understanding.

Major comments:

- It is unclear why the deconvolution of the data without considering a synchronization loss results in much stronger dynamics than considering a synchronization loss. On what basis did the authors

decide for 10% cell-cell variation? How would the data change if 5% or 20% cell-cell variation would be assumed?

- Why is aspartate labeling from glutamine (Figure S10) much less dynamic than nucleotide labeling from glutamine given that aspartate is needed for nucleotide production?

- The DCA data provided by the authors suggest PDK involvement in the cell cycle dependent regulation of glucose oxidation. To strengthen this mechanism PDK1-4 westernblots or PDH phosphorylation westernblots during the different cell cycle phases should be provided.

- Cells (such as HUH7 cancer cells) can grow without glutamine. It would be interesting to know whether these cells have a constant glucose oxidation rate throughout the cell cycle? I.e. do they have a constant lactate secretion and PDH phosphorylation?

- The changes in cell volume measured throughout the cell cycle should be provided as a Figure. Moreover, it should be discussed that the provided fluxes and dynamic data do not take compartment specific labeling and concentrations into account.

Minor:

- In the text it is stated that 10% cell-cell variation is assumed while in the Figures 11% is stated.

- Is the reductive glutamine flux a netflux to lipids (Metallo et al, Mullen et al) or a redox cycle (Jiang et al)?

- It should be discussed to which extent the results depend on the synchronization method.

-The following statement is confusing, since it is not clear how glycolytic flux was measured when not by glucose consumption.

"Notably, technical limitations with regard to the sensitivity of the LC MS analysis prevent the analysis of oscillations in glycolytic flux via direct measurements of changes in cell cycle dependent glucose consumption (considering that only ~1% of the glucose was consumed by the synchronized cell population in each three hour time interval). Still, our data show that the increase in glycolytic flux in G1/S phase co occurs with the increased glucose driven flux entering the TCA cycle."

Reviewer #2:

The manuscript studies the dynamic changes in metabolism that occur during cell cycling. The authors astutely point out several regulatory mechanisms that link cell cycle control to metabolic pathways - leading to the hypothesis that metabolic fluxes and metabolite levels are affected as cells progress through the G1, S and G2/M phases of the cell cycle. The authors have employ pulse-chase tracer experiments with stable-isotope labeled glucose and glutamine, along with ¹³C-based metabolic flux analysis (MFA) in synchronized HeLa cells to prove their hypothesis. In doing so, they present empirical data in the form of metabolite concentrations and isotopic enrichment, which show significant oscillatory traits. Most notably, their intracellular flux estimations also highlight the oscillatory nature of metabolism that follows the cell cycle. The novelty of their data lies in their quantitative observation of changes in glycolytic and TCA cycle fluxes as cells progress from G1 to S phase. Furthermore, the authors are able to provide strong evidence for why increased glutaminolysis and reduced glycolysis is essential for progression through the S phase. The other novel aspect of this manuscript lies in the usage of a modified Kinetic Flux Profiling (KFP) algorithm to estimate metabolic fluxes from pulse-chase experiment data where neither metabolic fluxes or isotopic enrichments are in steady state. I believe that the content of this manuscript is suitable for the high standard of publication of MSB, and will be of great interest to its readers. However, I strongly suggest the authors address the following issues before the manuscript can be accepted for publication.

Major Points

1. The use of average cell volume as a normalizer for flux estimations throughout the manuscript -

average volume is variable, in fact oscillating (Fig 1c). This makes comparison of "metabolic activity" or fluxes along the duration of the experiment complex. It's difficult for a reader to infer whether the oscillation is due to changes in metabolic reaction rates or changes in cell volume or both. I understand that using other conventional normalizers such cell number is tricky. However, I implore the authors to discuss in their manuscript on how to reconcile oscillatory behavior of these estimated fluxes vis-a-vis oscillating volume and/or oscillating reaction rates.

2. Page 3, Results, Paragraph 2, "We find that assuming 10% cell-cell variation ... synchronization dynamics". Here the authors are implying the assumed a value for intercellular variation, but, their methods section (Methods: Synchronization Model) describes that the σ was estimated to be 11% from the parameter-fitting model. Are the authors referring to different cell-cell variations? Please make this clear in the manuscript.

3. Page 5, Paragraph 3, "we measured the lactate concentrations in the culture media ...". The authors do not specify in the methods if the metabolite uptake and secretion rates were estimated in the same way as volume/isotopic enrichments were. Was the same spline-fitting method employed? To this end, the authors have not mentioned on which external fluxes (e.g. PDH flux, glutamine uptake, glutamate secretion) were constrained to measurements in the flux model

4.

a. Figure 6e,f. The authors mention the fraction of cells in G1, S and G2 phase, but, the units in the figures are clearly not fractions. I understand what the authors are trying to portray in the figure, however the way they have normalized the data to highlight the fractionation of cells among distinct phases might confuse some readers.

b. Page 7 last paragraph. Furthermore, the authors note that DCA treatment almost completely abolishes oscillations seen in metabolite concentrations and isotopic enrichment (Fig 6a-d). This is presumably due to DCA preventing cells to progress from S to G2. However, there only seems to be a 16% increase in cells in the S phase. This suggests that the efficiency of DCA isn't very high, but its effect on metabolism is significant. How do the authors reconcile these two observations?

Minor Points

1. Page 4, last paragraph, "we performed pulse chase ... every three hours for two cell cycles". Is there a reason for choosing 3-hour windows? Or was it arbitrarily chosen with the assumption that 3 hours would provide sufficient temporal resolution?

2. Page 5, first paragraph, "Specifically, we employed a variant of KFP ...". It would be helpful for the readers if the authors can include brief description in the corresponding methods section on (1) KFP and the (2) modifications/assumptions made to it for the purpose of this study.

3. Page 13, last paragraph. The authors have misspelled "time" as "tine".

Reviewer #3:

Summary

Performing metabolomics measurements in synchronized HeLa cell populations, Ahn et al. discovered the concentration of numerous intracellular metabolites to oscillate during the course of cell division. Due to the observed phase difference in the oscillating concentrations of key TCA metabolites (e.g. citrate and malate) the authors hypothesize a rewiring of the TCA cycle during the transition from the G1 to the S phase.

In order to confirm their hypothesis the authors performed pulse-trace experiments using uniformly labeled ^{13}C -glucose which feeds into the TCA cycle via pyruvate, and uniformly labeled ^{13}C -glutamine which feeds into the TCA cycle via α -ketoglutarate. From the intracellular ^{13}C labeling patterns of TCA metabolites, of nucleotides (pyrimidines), of amino-acids (aspartate), as well as of rates of lactate secretion, the authors reconstructed the glycolytic and TCA fluxes during cell cycle. For this reconstruction they used a computational method (Kinetic Flux Profiling).

The reconstructed flux distributions support the hypothesis that the TCA fluxes are rewired during cell cycle progression. In the late G1 phase, an increase in glycolysis, fuels the TCA cycle via pyruvate. In the S phase, the glycolytic flux is reduced, and TCA flux is compensated via reductive and oxidative glutamate metabolism. As shown here, glutamate metabolism in the S phase correlates with enhanced synthesis of pyrimidines, necessary for DNA replication, and contributes to a constant rate of NADH/FADH₂ production throughout the cell cycle.

The authors employ a dynamic perturbation experiment to support the importance of the evident TCA flux rewiring for cell cycle progression. When they chemically inhibit the pyruvate dehydrogenase kinase, an inhibitor of pyruvate dehydrogenase, thus forcing a high glycolysis, not only the metabolic oscillations stop, but also the cells cannot progress past the S phase.

Importantly, in order to get to these results the authors had to deconvolve the experimental data using mathematical models. These models helped them to correct for loss of synchrony of the cells, and volume changes of the cells.

General Remarks

The different metabolic requirements of the late G1 and the S phase have previously been described. Especially the work from Colombo et al. (2011 PNAS) which the authors rightfully cite in the submitted manuscript, it was shown that glucose is required for the G1/S transition, whereas glutamine is essential for the progression through the S phase. However, these conclusions were based on evidence of enzyme regulation (post-translational regulation of PFKFB3 and GLS1 by the Anaphase Promoting Complex) or extracellular fluxes (lactate secretion and glutamine consumption). Here, the authors move one step further, using state of the art methodologies to reconstruct intracellular TCA fluxes during the course of cell division. Thus, this work constitutes a conceptual advance, providing a detailed and dynamic approximation of the metabolic fluxes during cell division. Also, advanced computational methods are uniquely combined to analyze the experimental data.

Major concerns:

1) Unclear what the major contribution is: Is it a technical advance or does it does the work primarily advance our understanding about biology? The authors kept all the many technical things pretty much "under the hood". At some point I was asking myself what the major contribution of this manuscript would be: Would it be the biological insight on the intracellular fluxes? Or would it rather be the conceptual advance? Looking at the title that the authors have chosen, I guess, they consider both parts equally important. However, the cited Colombo paper to some extent already had uncovered the biology. Thus, I feel that the major novelty claim should come from the conceptual / methodological advance. In fact, in the very last paragraph they stress the novel "temporal-fluxomics approach". With regards to the methodological part, unfortunately, there is hardly anything described in the manuscript. Information is buried in the method section and also there the description is relatively short. I feel that much more weight should be given to the technical aspects of the work also in the main text. See also next point.

2) Testing assumptions is necessary: They authors use a number of assumptions and do not test the sensitivity of these assumptions towards the conclusions they draw from these analyses. I feel that the authors should thoroughly evaluate the assumptions underlying their method. In the discussion section, also the assumptions and the eventual limitations of the method should be discussed. I can envision that the authors could test assumptions through sensitivity studies or they could use independent measurements with which they benchmark the results they inferred from their computational analyses. In the following, please find a number of questions that I had with regards to assumptions. But note, in general, I was worried about the many assumptions and the points I mention in the following are not the only ones:

- a. what if during the cell cycle amino acid uptake / excretion would alter. How would this affect the conclusions?
- b. What is the effect of the metabolic network size used?
- c. A number of metabolites occur both in the cytoplasm and in the mitochondria. Do the author

consider these two compartments and the distribution of these metabolites across these compartments. If not, what kind of consequences does such a simplifying assumption have on the results?

d. The authors make a rather strong statement that there is a constant production rate of reducing equivalents. It is not precisely clear how they got to this and what kind of assumptions they authors used to get to this conclusion. If I look at the last paragraph of the method section, it seems like there were indeed several assumptions made. And then I get worried ...

3) Specific technical questions/concerns:

a. To estimate the intracellular metabolite concentration the authors divided the metabolite levels by the cell volume, which was corrected for loss of synchronization during the course of the experiment. This way, the concentrations of 57 metabolites were shown to exhibit significant oscillations, 44% percent of which could only be resolved after the *in silico* deconvolution of the cell volume dynamics. From this I understand that many metabolites the levels of which were found to oscillate, after the cell volume deconvolution their concentration would appear to be steady during the cell cycle. For instance, if the abundance of a metabolite increases in the G1 phase similarly to the deconvoluted cell volume, the concentration of the same metabolite (abundance / deconvoluted volume) will not oscillate. The authors may provide at least one example of such a metabolite.

b. During the DCA treatment the authors claim that glycolysis is upregulated. To support this claim they provide labeling data for citrate and acetyl-CoA from ¹³C-glucose and ¹³C-glutamate. These data show that not only metabolism stops oscillating after DCA addition, but also the glycolytic labeling is higher than the one derived from glutamate. In order, to substantiate their findings it is important that the lactate secretion rate is estimated after the addition of DCA. Given the partial (16%) arrest of the population in the S phase one would expect to see similarly higher lactate excretion rates on the population level after the addition of DCA.

1. The authors claim that the rewiring of the TCA cycle is important for progression through the S phase. Still, even though metabolism stops oscillating after the addition of DCA, only 16% of the population is arrested in the S phase. The remaining cells continue through the cell cycle even without metabolic oscillations. How do the authors explain that the majority of the cells divide even in the absence of an oscillating metabolism? The higher fraction of cells into the S phase after treatment with DCA could indicate partial arrest of the cell cycle, or increase in the duration of the S phase. The only way to resolve between the two is with single cell experiments (e.g. microscopy). Single cells may be treated with DCA and their cell cycle can be monitored. DNA stains other than PI, suitable for live imaging, may be applied to resolve the cell cycle phases.

2. The authors opted for a chemical inhibition of PDK to maintain PDH activity and increased glycolytic flux past the G1 phase. In fact, off-target effects have been reported for DCA (<https://www.ncbi.nlm.nih.gov/pmc/articles/PMC4622508/>). Given that I feel it is necessary to add at least one other perturbation experiment, for instance using an inhibitor of the mitochondrial pyruvate transporter.

3. Can it be excluded that the increase in glucose/glutamate levels (upon the addition of the labelled molecules) have any effect on cellular physiology?

4) Presentation of the results, the narration of the story: The core part of the manuscript is primarily a description of the results shown in the figures. As such, the manuscript is rather boring to read. I would prefer seeing that the authors would take the reader also through the journey of their technical developments (see also point 1), provide the reader with analyses of the many assumptions and a more thorough validation of the obtained results (see point 2).

Minor Comments

1. When comparing HeLa to HCT-116 cells, the latter are arrested both in the S and the G2 after DCA treatment. Given the different physiology of the two cellular types, can you explain this finding?

2. In the main text it is not clear how the cell cycle phases were resolved. Refer to the PI measurements inside the main text and in the legend of Fig. 1A.

3. In Fig. S1, the plots corresponding to 3h and 9h, as well as 6h and 12h are identical.

4. In Fig. S2 the authors plot the ATP/ADP ratio. The ATP concentration is shown to oscillate in the main text Fig. 2. However, there is no evidence that ADP oscillates. Are the dynamics of the ATP concentration different than the ATP/ADP ratio?

5. At the synchronization loss model please include units for the parameters and variables. For example, γ is a rate expressed as cell cycles per time unit. When multiplied by time, then it is a measure of the progression into the cell cycle (e.g. 0.5 for half-way through the cell cycle). The cell doubling time is d_{dc} expressed in time units and the duration of each cell cycle phase $dG1$, dS and dGM .
6. What kind of distribution is this in Eq II?
7. At the synchronization loss model description, the durations of the cell cycle phases are mentioned ($dG1 = 6.8h$, $dS = 6.4h$ and $dGM = 3.2h$). If the G1 and S phases had the same duration, they should be represented equally into the unsynchronized population. However this is not the case as we can see in Fig. 1A.
8. Right y axis units in Fig. 1A missing.
9. Include also dTTP data in Fig. 2 (from Fig. S11).
10. It is interesting that although pyrimidines peak in the S phase (their synthesis is boosted by glutamate influx into the TCA cycle), the purines do not oscillate in phase. In fact dATP peaks in the S phase, in phase with dTTP and dCTP, but dGTP peaks in the G1 phase. Please comment on this.
11. Fig. S4: there is not black-dashed line.
12. In the computational deconvolution of metabolite concentration is $u_i(t)$ estimated using the measured or the corrected volume?
13. In Fig. 1C, how was the green line (volume in case of no synchronization loss) determined?
14. Blend figures 3b and S3. In fact, it is confusing when different nonmeclatures are used to describe reactions in the network. Also, with all the labeling patters (e.g. $m+2$, $m+3$, $m+4$, $m+5$) present all around in the text, it would be nice if those terms were also included in Fig. 3b.
15. Assumptions of the model: Influence of the applied network size? Mitochondrial volume? Metabolites originate from both the cytoplasm and the mitochondrion. What kind of consequences does this have?
16. P. 5. 3rd paragraph: 30% increase in lactate secretion rate. How was lactate measured in the extracellular environment. I might have overlooked it, but I feel that the authors did not mention how they measured extracellular metabolites. Also, how from such extracellular metabolite concentration measurements did they estimates rates?
17. Method section on LC-based metabolomics: It is stated that conditioned medium was used. Please provide more detail, such that others can in fact reproduce this experiment.
18. Page 15: second last line: Grammar?
19. Optimizations were started multiple times, but the authors did not say how often they in fact started the optimizations.

1st Revision - authors' response

22 August 2017

We would like to thank the reviewers for the constructive comments on our manuscript. We have addressed all comments as detailed below in this revised version.

Reviewer #1:

Major comments:

- It is unclear why the deconvolution of the data without considering a synchronization loss results in much stronger dynamics than considering a synchronization loss. On what basis did the authors decide for 10% cell-cell variation? How would the data change if 5% or 20% cell-cell variation would be assumed?

Given the concentration measurements of a certain metabolite throughout the cell cycle, we utilize the synchronization loss model to compute the expected dynamics in case of no synchronization loss in the cell population (i.e. the deconvoluted signal; $u_i'(x)$ in Eq. 7). E.g. this is shown in Figure 1d for CTP, where the measured concentrations throughout the cell cycle are shown in red, and the deconvoluted signal shown in green. We further show a simulation of the expected concentration of CTP in cells that loose synchronization based on the inferred

deconvoluted concentrations dynamics (i.e. the convoluted signal; based on Eq. 7) that is shown to match the actual experimental measurements. The amplitude of the oscillations in the convoluted metabolite concentration dynamics is not larger than that in the deconvoluted signal (as the convoluted signal represents averaging of the true signal with respect to the distribution of cell-intrinsic times; Eq. 7). The caption in Figure 1d was modified to better explain the above.

The extent of cell-cell variability in the progression rate through the cell cycle was determined by optimizing the parameters of the “synchronization loss model”. Specifically, we perform a maximum log-likelihood estimation of the four parameters of the model (σ , d_G , d_S , and d_M), minimizing the variance-weighted sum of squared residuals between the simulated number of cells in the different cell cycle phases ($m_S(t)$, $m_G(t)$ and $m_M(t)$) and the FACS measurements. We found that cell-cell variability of 11% provides an optimal fit between the model simulations and measurements (shown in Figure 1b). In the previous version of the manuscript, the 11% was indicated in the caption of Figure 1 and Methods, while it was specified to be 10% in the main text by mistake. Following the reviewer’s comment, the main text was modified to indicate that the 11% cell-cell variation was derived as explained in the Methods and here. We further elaborate on the maximum-likelihood estimation of the model parameters in the Methods.

Following the reviewer’s comment, we now also compute a confidence interval for each of the “synchronization loss model” parameters, evaluating the quality of the fit between the simulated and measured FACS data for a range of possible values per each parameter. We find that with 95% probability, the cell-cell variability is between 10% and 12% (see Methods).

- Why is aspartate labeling from glutamine (Figure S10) much less dynamic than nucleotide labeling from glutamine given that aspartate is needed for nucleotide production?

The fact that aspartate is indeed a precursor for pyrimidine biosynthesis entails that the fraction of the aspartate pool being labeled after 1 hour feeding with ^{13}C -glutamine (at different times throughout the cell cycle) must be higher than that of pyrimidines; and indeed the relative abundance of aspartate $m+4$ is ~ 0.7 throughout the cell cycle (Figure S9), while that of CTP $m+3$ oscillates around ~ 0.05 (Figure 4e).

The amplitude of the oscillation is indeed markedly higher for CTP (changing ~ 2 -fold between G1 and S) than for aspartate (with barely detectable oscillations in $m+4$ labeling). The size of the oscillation in the labeling of each metabolite is affected by several factors, including the oscillation in the concentration of the metabolite, the oscillation in the labeling of its precursors, and the change in its biosynthetic flux. Specifically, the large oscillation in the labeling of CTP $m+3$ is a result of large oscillations in the concentration of CTP (Figure 2) and flux towards pyrimidine biosynthesis (Figure 5g); low CTP $m+3$ appears in G1 (where CTP concentration is high and the pyrimidine biosynthetic flux is low), and high CTP $m+3$ appears in S (where CTP concentration is low and pyrimidine biosynthesis flux is high).

- The DCA data provided by the authors suggest PDK involvement in the cell cycle dependent regulation of glucose oxidation. To strengthen this mechanism PDK1-4 westernblots or PDH phosphorylation westernblots during the different cell cycle phases should be provided.

We thank the reviewer for this very helpful suggestion. Following the reviewer’s comment, we measured the phosphorylation of PDH throughout the cell cycle in HeLa cells, considering three known phosphosites: Ser293, Ser300, and Ser232. Phosphorylation of Ser293 was measured via Western Blotting, while that of Ser300 and Ser232 via commercially available ELISA kits (Abcam). We find a significant increase in phosphorylation of Ser232 in S phase (now included as Figure S19; see below), consistent with the decrease in PDH flux observed in S phase. The phosphorylation in other phosphosites does not change significantly throughout the cell cycle; also the concentration of PDH remains constant throughout the cell cycle. Notably, a previous study has noted a translocation of the PDH complex from mitochondria to the nucleus in the S phase of the HeLa cells (Sutendra et al, Cell, 2014), suggesting a drop in PDH concentration specifically in mitochondria in S phase. Further studies are required to determine the precise regulatory mechanism that underlies the cell cycle oscillations in glycolytic flux into TCA cycle reported here, which may potential spread quantitatively among several enzymes (in accordance

with the view of Metabolic Control Analysis (Fell, 1997)) – e.g. via the control of pyruvate transport into mitochondria or of citrate synthase.

Figure S19: (a) Phosphorylation of PDH Ser232 increases in S phase. HeLa cells were synchronized using double thymidine block and lysates were prepared at different stages of the cell cycle. PDH phosphorylation was measured via an ELISA kit. All experiments were done in triplicates; two tailed Student's T-test was applied to calculate p-values. Notably, the concentration of PDH (measured via an ELISA kit) remains constant throughout the cell cycle. (b) DCA treatment significantly decreases PDH phosphorylation at Ser232.

- Cells (such as HUH7 cancer cells) can grow without glutamine. It would be interesting to know whether these cells have a constant glucose oxidation rate throughout the cell cycle? I.e. do they have a constant lactate secretion and PDH phosphorylation?

The paper presents a new methodology for analyzing metabolic flux dynamics throughout the cell cycle and the application of this approach to HeLa cells – a most extensively studied cell line, which is commonly used as a testbed for new 'omics' techniques. We expect future studies to apply this approach to investigate metabolic oscillations in variety of cancer cell lines under diverse physiological conditions.

- The changes in cell volume measured throughout the cell cycle should be provided as a Figure. Moreover, it should be discussed that the provided fluxes and dynamic data do not take compartment specific labeling and concentrations into account.

The oscillations in cell volume measured throughout cell cycle as well as the deconvoluted signal (i.e. average cell volume expected in case of no synchronization loss) are shown in Figure 1c. We now added a paragraph in the Discussion specifying that our analysis do not take into account potential differences in the concentration and labeling dynamics of metabolites in distinct subcellular compartments (as typically performed with Metabolic Flux Analysis). Notably, the presented temporal-fluxomics approach can accommodate such compartment-specific measurements if/when available.

Minor:

- In the text it is stated that 10% cell-cell variation is assumed while in the Figures 11% is stated.

The text was modified to indicate 11%.

- Is the reductive glutamine flux a netflux to lipids (Metallo et al, Mullen et al) or a redox cycle (Jiang et al)?

The oxidative flux through isocitrate dehydrogenase was found to be higher than the corresponding reductive flux all throughout the cell cycle (in agreement with Fan et al., JBC 2013). And hence, under the studied conditions, the net flux through this enzyme is in the oxidative direction (as also mentioned in the Discussion).

- It should be discussed to which extent the results depend on the synchronization method.

A major emphasis in this work has been given to minimize the perturbation induced by the specific choice of synchronization method – i.e. following cells for almost 3 complete cell cycles and starting the metabolomics studies after giving cells sufficient time to complete one cell cycle and recover from a potential perturbation etc. This is specified in the main text and in the Discussion. In the revised version, we also compare the durations of each cell cycle phase inferred based on optimizing the parameters of the “synchronization loss model” to those inferred based on PI staining/FACS analysis of non-synchronized cells. A small under estimation of the duration of G1 and over estimation of the duration of S by the analysis of the synchronized cell (both not more than 1h off the measurements in the non-synchronized cells) is attributed to a potential slight perturbation to cell cycle dynamics due to synchronization-induced growth arrest.

-The following statement is confusing, since it is not clear how glycolytic flux was measured when not by glucose consumption. "Notably, technical limitations with regard to the sensitivity of the LC MS analysis prevent the analysis of oscillations in glycolytic flux via direct measurements of changes in cell cycle dependent glucose consumption (considering that only ~1% of the glucose was consumed by the synchronized cell population in each three hour time interval). Still, our data show that the increase in glycolytic flux in G1/S phase co occurs with the increased glucose driven flux entering the TCA cycle."

Glycolytic flux was quantified based on measuring oscillations in the secretion rate of lactate throughout the cell cycle (while showing that fluxes through pathways that branch out of glycolysis are at least two orders of magnitude smaller; and ~99% of the carbons in lactate originate from glucose in these cells). Following the reviewer’s comment, we further elaborate in the Methods regarding the approach used to perform deconvolution of lactate secretion measurements and estimate the dynamics of lactate secretion throughout the cell cycle (see also reply to comment #3 by the 2nd reviewer below). We further rephrased the text as follows: “Notably, analyzing oscillations in glycolytic flux based on direct measurements of changes in glucose consumption throughout the cell cycle (rather than based on lactate secretion and flux through branching pathways as described above) was not possible due to technical difficulty in accurately quantifying glucose consumption by synchronized cells within 3 hour time intervals (considering that the synchronized cells consume ~1% of the glucose in media within this short time period).”

Reviewer #2:

Major Points

1. The use of average cell volume as a normalizer for flux estimations throughout the manuscript - average volume is variable, in fact oscillating (Fig 1c). This makes comparison of "metabolic activity" or fluxes along the duration of the experiment complex. It's difficult for a reader to infer whether the oscillation is due to changes in metabolic reaction rates or changes in cell volume or both. I understand that using other conventional normalizers such cell number is tricky. However, I implore the authors to discuss in their manuscript on how to reconcile oscillatory behavior of these estimated fluxes vis-a-vis oscillating volume and/or oscillating reaction rates.

*All flux estimations described in the paper are in fmole/(pL-cells * h) – i.e. , mM/h. Multiplying a flux estimate in time x within the cell cycle with the estimated cell volume at that time (in pL, as shown in Figure 1c; $v'(x)$ in Equation 6) gives a flux per cell – i.e. in fmole/(cell*h). This is now explicitly specified in the paper.*

2. Page 3, Results, Paragraph 2, "We find that assuming 10% cell-cell variation ... synchronization dynamics". Here the authors are implying the assumed a value for intercellular variation, but, their methods section (Methods: Synchronization Model) describes that the σ was estimated to be 11% from the parameter-fitting model. Are the authors referring to different cell-cell variations? Please make this clear in the manuscript.

The cell-cell variation in progression rate throughout the cell cycle was indeed inferred using parameter-fitting, as explained in the Methods. The inferred 11% variation was indicated in caption of Figure 1 and in Methods, while by mistake indicated as 10% in the main text – we thank the reviewer for pointing this out. This is now fixed. Also, following the reviewer's comment, we modified the main text to better explain how this value was derived: "The parameters of the model were estimated by fitting a simulation of how the synchronized cell population progresses through the different phases of the cell cycle with corresponding FACS measurements, finding that cell-cell variability in the rate of cell cycle progression through the cell cycle is 11% (Figure 1b; Methods)."

3. Page 5, Paragraph 3, "we measured the lactate concentrations in the culture media ...". The authors do not specify in the methods if the metabolite uptake and secretion rates were estimated in the same way as volume/isotopic enrichments were. Was the same spline-fitting method employed? To this end, the authors have not mentioned on which external fluxes (e.g. PDH flux, glutamine uptake, glutamate secretion) were constrained to measurements in the flux model

We thank the reviewer for pointing this out. We now added a detailed description of how lactate and glutamate secretion rates throughout the cell cycle were derived, combining LC-MS based measurement of these metabolites in media samples of synchronized cells, followed by computational deconvolution (via a conceptually similar spline-fitting approach used for the deconvolution of volume, intracellular concentrations, and isotopic labelling measurements; now described in details to the Methods section):

We denote the measured change in pool size of metabolite i in the culture media between time $t - \Delta t$ and time t by $\Delta e_i(t)$. We estimate the transport flux of metabolite i by the synchronized cell population at time t, denoted $f_i(t)$ (in molar amount per unit of cell volume per hour; with positive and negative flux representing secretion and uptake, respectively) by dividing the change in pool size of metabolite i by the accumulated volume of cells in the culture metabolite between time $t - \Delta t$ and time t:

$$f_i(t) = \frac{1}{\int_{t-\Delta t}^t \int_0^{d_{CYC}} v'(x)g(x,t')dx dt'} \Delta e_i(t) \quad , \quad (\text{Eq. 9})$$

The transport flux of metabolite i at cell-intrinsic time x, denoted $f'_i(x)$ is estimated as:

$$\Delta e_i(t) = \int_{t-\Delta t}^t \int_0^{d_{CYC}} f'_i(x) v'(x)g(x,t')dx dt' + \varepsilon(t), \quad (\text{Eq. 10})$$

considering that the measured change in pool size of metabolite i at time t represents the cumulative transport within the Δt time interval by cells with different intrinsic cell cycle time x .

4. a. Figure 6e,f. The authors mention the fraction of cells in G1, S and G2 phase, but, the units in the figures are clearly not fractions. I understand what the authors are trying to portray in the figure, however the way they have normalized the data to highlight the fractionation of cells among distinct phases might confuse some readers.

We thank the reviewer for pointing this out. The label on the y-axis is now changed to "Number of cells after DCA treatment over control".

b. Page 7 last paragraph. Furthermore, the authors note that DCA treatment almost completely abolishes oscillations seen in metabolite concentrations and isotopic enrichment (Fig 6a-d). This is presumably due to DCA preventing cells to progress from S to G2. However, there only seems to be a 16% increase in cells in the S phase. This suggests that the efficiency of DCA isn't very high, but its effect on metabolism is significant. How do the authors reconcile these two observations?

The 16% increase in the number of cells in S phase is after three hours treatment of HeLa cells with DCA. Feeding HeLa cells for a longer time lead to a more substantial increase in the fraction of cells in S phase; as shown in Figure 6f, where feeding non-synchronized HeLa cells with DCA for 24h leads to a ~30% increase in the fraction of cells in S phase. Notably, following a comment raised by referee #3, additional experiments were performed showing that DCA treatment dose not arrest cells in S phase but rather prolong the duration of S phase (see below). Overall, our results show that a shift from glucose to glutamine-derived flux into TCA cycle plays an important role in cellular progression through S phase.

Minor Points

1. Page 4, last paragraph, "we performed pulse chase ... every three hours for two cell cycles". Is there a reason for choosing 3-hour windows? Or was it arbitrarily chosen with the assumption that 3 hours would provide sufficient temporal resolution?

The 3-hour time window was selected such that we obtain at least 2 metabolomics measurements in G1 and S (whose length is >6h). Going below 3 hour windows and performing pulse change isotope tracing for 45 hours would be technically hectic.

2. Page 5, first paragraph, "Specifically, we employed a variant of KFP ...". It would be helpful for the readers if the authors can include brief description in the corresponding methods section on (1) KFP and the (2) modifications/assumptions made to it for the purpose of this study.

Following the reviewer's comment (as well as comment made by the 3rd reviewer regarding giving more weight to technical aspects in the main text), we added the following overview of the employed computational approach in the Results section: "The inferred oscillations in metabolite isotopic labeling and concentrations were used to computationally analyze metabolic flux dynamics throughout the cell cycle, utilizing a variant of Kinetic Flux Profiling (KFP) (Yuan et al, 2008) (Figure 5; in units of nmole/uL-cells/h; i.e., mM/h; Methods). Specifically, given a metabolite whose isotopic labeling dynamics throughout the cell cycle was inferred as explained above, we search for the most likely transient production and consumption fluxes in each one-hour interval through the cell cycle, such that the simulated labeling kinetics of this metabolite (within the one-hour interval) would optimally match the experimental measurements (Methods; Figure S3-8). The simulation of the isotopic labeling kinetics of a metabolite of interest within a one-hour time interval is performed via an ordinary differential equations (ODE) model; relying on the inferred concentration of the metabolite within this time-interval (considering that a metabolite with a larger pool size would take more time to label, per unit of flux), as well as the isotopic labeling kinetics of intermediates that produce this metabolite. While KFP is typically applied to estimate fluxes under metabolic steady state (in which fluxes satisfy a stoichiometric mass-balance constraint), here, we constrain the difference between transient fluxes that

produce and consume a certain metabolite according to the measured momentary change in the concentration of that metabolite.”

3. Page 13, last paragraph. The authors have misspelled "time" as "tine".

Fixed.

Reviewer #3:

Major concerns:

1) Unclear what the major contribution is: Is it a technical advance or does it does the work primarily advance our understanding about biology? The authors kept all the many technical things pretty much "under the hood". At some point I was asking myself what the major contribution of this manuscript would be: Would it be the biological insight on the intracellular fluxes? Or would it rather be the conceptual advance? Looking at the title that the authors have chosen, I guess, the consider both parts equally important. However, the cited Colombo paper to some extent already had uncovered the biology. Thus, I feel that the major novelty claim should come from the conceptual / methodological advance. In fact, in the very last paragraph they stress the novel "temporal-fluxomics approach". With regards to the methodological part, unfortunately, there is hardly anything described in the manuscript. Information is buried in the method section and also there the description is relatively short. I feel that much more weight should be given to the technical aspects of the work also in the main text. See also next point.

In our view, the methodological and biological contributions in this manuscript are equally important.

*In terms of biological contribution, the paper by Colombo et al (as well as other related publications) shows cell cycle dependent changes in the concentration of proteins involved in glycolysis and glutaminolysis (as well as some measurement of cell cycle dependent changes in lactate/glutamate secretion rates). However, deriving the first quantitative view of intracellular metabolic flux oscillations in a human cell line, our manuscript provides several important biological contributions: (i) we show the increase in glycolytic flux in G1/S is further associated with an increase in glucose oxidation in TCA cycle; (ii) we show that the increase in glutamine consumption in S phase results in induced oxidative and reductive TCA cycle flux and support nucleotide biosynthesis, involving oscillations in flux through a variety of enzymes. (iii) We show that the complementary flux oscillations between glucose and glutamine oxidation in TCA cycle maintain a constant production rate of reducing equivalents and oxidative phosphorylation flux throughout the cell cycle. (iv) We show that the shift from glucose to glutamine oxidation in S phase is important for cell cycle progression and cell proliferation. **To the best of our knowledge, this is the first report of oscillations in TCA cycle flux throughout the mammalian cell cycle.***

In terms of methodological contribution, following the reviewer's comment, we now further elaborate on the computational flux modelling and deconvolution approaches in the main text: We added a detailed description of the deconvolution of measurements of metabolite secretion fluxes to the culture media. We now further elaborate in the main text on the computational approach used for the estimation of transient fluxes by integrating measurements of oscillations in metabolite concentration and isotopic labeling throughout the cell cycle.

2) Testing assumptions is necessary: They authors use a number of assumptions and do not test the sensitivity of these assumptions towards the conclusions they draw from these analyses. I feel that the authors should thoroughly evaluate the assumptions underlying their method. In the discussion section, also the assumptions and the eventual limitations of the method should be discussed. I can envision that the authors could test assumptions through sensitivity studies or they could use independent measurements with which they benchmark the results they inferred from their computational analyses. In the following, please find a number of questions that I had with regards to assumptions. But note, in general, I was worried about the many assumptions and the points I mention in the following are not the only ones:

a. what if during the cell cycle amino acid uptake / excretion would alter. How would this affect the conclusions?

Theoretically, oscillations in the uptake rate of glucogenic and ketogenic amino acids throughout the cell cycle could indeed bias the TCA cycle flux estimations. However, feeding non-synchronized HeLa cells with [U-¹³C]-glucose and [U-¹³C]-glutamine for 24h shows that

more than 97% of the carbons in TCA cycle metabolites (citrate, malate, and aKG) are derived from these substrates or from CO_2 . Hence, potential oscillations in amino acid catabolic fluxes would have little or no effect on the reported flux oscillations. This is now referred to in the Discussion and shown in Figure S17 (also shown here).

Figure S17: Fraction of carbons labeled in TCA cycle intermediates and other metabolites when feeding both $[U-^{13}C]$ -glucose and $[U-^{13}C]$ -glutamine for 24h. Additional carbons are derived from the fixation of atmospheric CO_2 through reductive isocitrate dehydrogenase (IDH) and pyruvate carboxylase (PC). E.g. for citrate, the fraction of citrate m+5 when feeding isotopic glutamine (as an indication of reductive IDH activity) and malate m+3 when feeding isotopic glucose (as an indication pyruvate carboxylase activity) suggest that another ~3% of the citrate carbons are derived from CO_2 fixation.

b. What is the effect of the metabolic network size used?

Not accounting for potential oscillations through other reactions implicated in central metabolism of glucose and glutamine could in principle also bias the presented flux estimations (though this is the common practice with Metabolic Flux Analysis, due to an implicit assumption regarding the considered reactions being the most important/high flux reactions in central metabolism; and experimental difficulty in measuring the isotopic labeling dynamics and concentration of all related metabolites). Notably, the flux analysis here is performed separately through groups of converging reactions producing different metabolites (e.g. citrate synthase and reductive IDH producing citrate; and α -ketoglutarate oxidation and pyruvate carboxylase producing malate/oxaloacetate; etc). Hence a potential bias in some of these independent flux estimations would not change the overall emerging view of glucose and glutamine oscillations; e.g. the increased TCA cycle metabolism of glutamine in S phase is supported by several independent flux estimations showing increased glutamine-derived flux into TCA cycle in S phase, increased oxidation of α -ketoglutarate, increased reduction of α -ketoglutarate, increased malic enzyme flux, and increased nucleotide biosynthesis. We elaborate on the above in the Discussion.

c. A number of metabolites occur both in the cytoplasm and in the mitochondria. Do the author consider these two compartments and the distribution of these metabolites across these compartments. If not, what kind of consequences does such a simplifying assumption have on the results?

A simplifying assumption that facilitated the estimation in flux dynamics through the cell cycle is of rapid mixing between mitochondrial and cytosolic metabolite pools, estimating whole-cell level fluxes. This assumption is typically made when analyzing flux in eukaryotic cells due to experimental difficulty in measuring metabolite concentrations and labelling dynamics in distinct subcellular compartments. Methodological advancement in metabolomics measurements in distinct subcellular compartments would enable further studies on cell cycle oscillations in metabolic flux in mitochondria versus cytosol. This is now stated in the Discussion.

d. The authors make a rather strong statement that there is a constant production rate of reducing equivalents. It is not precisely clear how they got to this and what kind of assumptions they authors used to get to this conclusion. If I look at the last paragraph of the method section, it seems like there were indeed several assumptions made. And then I get worried ...

The rate of NADH/FADH₂ production from the oxidation of glucose and glutamine at each time point throughout the cell cycle is calculated by summing up the flux through all NADH and FADH₂ producing reactions in the model. As mentioned above, the catabolism of glucose and glutamine accounts for more than 97% of the flux through the considered reactions. We find that indeed, the total rate of NADH/FADH₂ production remains pretty constant through the cell cycle (Figure 5e). In support of this intriguing finding, we show that oxygen consumption used for oxidative phosphorylation (measured using a Seahorse XFp Flux Analyzer; the non-mitochondrial oxygen consumption after treatment with the ETC inhibitors rotenone and antimycin A subtracted from the basal OCR) remains constant throughout the cell cycle.

The assumptions described in the Methods refer to how we split the total NADH/FADH₂ production to the part derived from glucose oxidation versus that from glutamine oxidation. Following the reviewer's comment, we elaborate on the experimental measurements that underlie these assumptions (see Methods).

3) Specific technical questions/concerns:

a. To estimate the intracellular metabolite concentration the authors divided the metabolite levels by the cell volume, which was corrected for loss of synchronization during the course of the experiment. This way, the concentrations of 57 metabolites were shown to exhibit significant oscillations, 44% percent of which could only be resolved after the in silico deconvolution of the cell volume dynamics. From this I understand that many metabolites the levels of which were found to oscillate, after the cell volume deconvolution their concentration would appear to be steady during the cell cycle. For instance, if the abundance of a metabolite increases in the G1 phase similarly to the deconvoluted cell volume, the concentration of the same metabolite (abundance / deconvoluted volume) will not oscillate. The authors may provide at least one example of such a metabolite.

To estimate the intracellular concentration of a metabolite, we divide the measured metabolite pool size in the synchronized cell population by the measured total-cell volume of the population, rather than by the deconvoluted volume (deriving $u_i(t)$ in Equation 7; this has now been made clearer in the Methods). The deconvolution of this concentration estimation based on Equation 7 makes usage of previously deconvoluted volume of cells in each intrinsic time x within the cell cycle (based on Equation 6), and to fraction of cells in the population with intrinsic time x .

While the entire paper describes metabolite concentrations, metabolite uptake, and fluxes per unit of cell volume, inferred quantities can be easily transformed to per-cell: Dividing the inferred metabolite concentration of metabolite i at intrinsic time x ($u'_i(x)$) by the average volume of a cell in intrinsic time x ($v'(x)$) would give the pool size of the metabolite per cell. We refer to this unit transformation of inferred fluxes in the Methods.

b. During the DCA treatment the authors claim that glycolysis is upregulated. To support this claim they provide labeling data for citrate and acetyl-CoA from ¹³C-glucose and ¹³C-glutamate. These data show that not only metabolism stops oscillating after DCA addition, but also the glycolytic labeling is higher than the one derived from glutamate. In order, to substantiate their findings it is important that the lactate secretion rate is estimated after the addition of DCA. Given the partial (16%) arrest of the population in the S phase one would expect to see similarly higher lactate excretion rates on the population level after the addition of DCA.

DCA treatment inhibits PDKs (which negatively regulate PDH) and hence activates PDH – increasing glycolytic flux entering TCA cycle (as previously reported; and as shown here). Diverting glycolytic flux into TCA cycle, DCA treatment is expected to significantly lower lactic acid secretion to the medium; and, in fact, DCA was tested as a drug for patients with lactic

acidosis as a mean to lower lactic acid excretion (Blackshear PJ et al., *Diabetes Care*. 1982). Indeed, applying metabolomics to media samples of non-synchronized HeLa and HCT-116 cells shows a significant drop in lactic acid secretion rate after DCA treatment (see Figure below).

Lactate secretion rate in HeLa and HCT-116 upon 16mM DCA treatment for 24h.

1. The authors claim that the rewiring of the TCA cycle is important for progression through the S phase. Still, even though metabolism stops oscillating after the addition of DCA, only 16% of the population is arrested in the S phase. The remaining cells continue through the cell cycle even without metabolic oscillations. How do the authors explain that the majority of the cells divide even in the absence of an oscillating metabolism? The higher fraction of cells into the S phase after treatment with DCA could indicate partial arrest of the cell cycle, or increase in the duration of the S phase. The only way to resolve between the two is with single cell experiments (e.g. microscopy). Single cells may be treated with DCA and their cell cycle can be monitored. DNA stains other than PI, suitable for live imaging, may be applied to resolve the cell cycle phases.

We thank the reviewer for this helpful suggestion. Following the reviewer's comment, we further investigated the effect of DCA treatment on cell cycle progression, aiming to differentiate between prolonging the time cells spend in S phase and S phase arrest. Towards this end, we fed HeLa cells with the fluorescent cell proliferation marker CFSE (carboxyfluorescein succinimidyl ester) and measured the fraction of cells that continue to proliferate upon DCA treatment. CFSE binds cellular proteins and its fluorescence is retained for 7-8 generations, while the signal drops after each mitotic event. We find that feeding DCA for 72 hours, almost all cells (~97%) complete at least one cell cycle (now shown in Figure S15 and here below). This suggests that DCA treatment and corresponding elimination of oscillations in glycolytic flux into TCA cycle do not arrest cells in S phase. However, as indicated in the main text, DCA treatment significantly increases the relative time cells spend in S phase, by 30% and 17% in HeLa and HCT-116, respectively – suggesting that the shift from glucose to glutamine-derived flux into TCA cycle plays an important role in cellular progression through S phase. This is now described in the last subsection in the Results.

Figure S15: FACS measurement of CFSE signal in non-synchronized HeLa cells immediately after feeding to cells (a) and after 72 hours (b).

2. The authors opted for a chemical inhibition of PDK to maintain PDH activity and increased glycolytic flux past the G1 phase. In fact, off-target effects have been reported for DCA (<https://www.ncbi.nlm.nih.gov/pmc/articles/PMC4622508/>). Given that I feel it is necessary to add at least one other perturbation experiment, for instance using an inhibitor of the mitochondrial pyruvate transporter.

Following the reviewer's comment, we tested the effect of treating HeLa cells with a mitochondrial pyruvate carrier inhibitor (UK5099) on cell cycle progression. Expectedly, treating the cells with 200 μ M UK5099 for 1 hr leads to a major drop in glycolytic flux into TCA cycle, as evident by the increase in citrate m+5/m+4 ratio from [U-¹³C]-glutamine (now shown in Figure S16a; and below). This is the opposite effect to DCA treatment, inducing PDH and hence increasing glycolytic flux into TCA cycle (decreasing the citrate m+5/m+4 ratio). And accordingly, while DCA treatment was shown in the paper to prolong S phase and increase the fraction of cells in this phase (inducing glycolytic flux into TCA cycle when it is low), UK5099 treatment had the opposite effect of decreasing the fraction of cells in S phase. We now refer to this in the main text and in Figure S16.

Figure S16: (a) Citrate m+5/m+4 ratio when feeding non-synchronized HeLa cells with an inhibitor of the mitochondrial pyruvate transporter (UK5099) versus when feeding DCA. (b) The fraction of cells in G1, S, and G2/M phases in non-synchronized HeLa cells after 24 hour treatment with UK5099 (normalized by measurements in untreated control cells).

3. Can it be excluded that the increase in glucose/glutamate levels (upon the addition of the labelled molecules) have any effect on cellular physiology?

The pulse-chase labeling experiments were performed by aspirating the media and providing

conditioned media with the same concentration of isotopic glucose/glutamine as before. Hence, cells should not experience a change in glucose/glutamine concentration. Notably though, the concentration of glucose/glutamine in the media of the synchronized cells does drop within the 45 hours in which they are grown. Aiming to overcome that, we use high concentration of both glucose (25mM) and glutamine (3mM) to begin with, to minimize their depletion (and indeed the drop in glucose concentration was less than 10% and that of glutamine less than 40% after 45 hours).

4) Presentation of the results, the narration of the story: The core part of the manuscript is primarily a description of the results shown in the figures. As such, the manuscript is rather boring to read. I would prefer seeing that the authors would take the reader also through the journey of their technical developments (see also point 1), provide the reader with analyses of the many assumptions and a more thorough validation of the obtained results (see point 2).

Following the reviewer's comment (and related comments by the other reviewers), we now elaborate in the Results on technical aspects regarding the various deconvolution methods as well as on the employed flux analysis methodology. We now elaborate on the underlying assumption and potential limitations of the temporal-fluxomics approach, including choice of network size, potential contribution of carbon sources other than glucose and glutamine, and on subcellular-compartmentalization issues in the Discussion.

Minor Comments

1. When comparing HeLa to HCT-116 cells, the latter are arrested both in the S and the G2 after DCA treatment. Given the different physiology of the two cellular types, can you explain this finding?

We currently cannot explain why DCA treatment leads to accumulation of cells not only in S phase but also in G2 in HCT-116.

2. In the main text it is not clear how the cell cycle phases were resolved. Refer to the PI measurements inside the main text and in the legend of Fig. 1A.

We now refer to the PI staining/FACS analysis in the main text and figure caption.

3. In Fig. S1, the plots corresponding to 3h and 9h, as well as 6h and 12h are identical.

Figure S1 was corrected. We thank the reviewer for pointing our attention to this mistake.

4. In Fig. S2 the authors plot the ATP/ADP ratio. The ATP concentration is shown to oscillate in the main text Fig. 2. However, there is no evidence that ADP oscillates. Are the dynamics of the ATP concentration different than the ATP/ADP ratio?

ADP is not significantly oscillating and that is why it does not appear in Figure 2. The dynamics of ATP/ADP ratio are indeed quite similar to those of ATP (peaking in G1/S).

5. At the synchronization loss model please include units for the parameters and variables. For example, γ is a rate expressed as cell cycles per time unit. When multiplied by time, then it is a measure of the progression into the cell cycle (e.g. 0.5 for half-way through the cell cycle). The cell doubling time is d_{cdc} expressed in time units and the duration of each cell cycle phase d_{G1} , d_S and d_{GM} .

We now indicate that the units for d_{CYC} , d_G , d_S , and d_M are hours. The relative progression rate of a cell through the cell cycle, γ , represents a ratio between the intrinsic time of a cell and the real time (and hence has no units). We added the following text to better explain the meaning of γ : "E.g. for a cell with a relative progression rate through the cell cycle of $\gamma=1$ released from growth arrest it will take d_S+d_M hours to complete one cell cycle (and then have an intrinsic

time of $x=0$), while for a cell with double the rate of $\gamma=2$ it will take half the time (i.e. $(d_S+d_M)/2$)."

6. What kind of distribution is this in Eq II?

We now elaborate on how the distribution $g(x, t)$ is defined: "At time t post the release from synchronization arrest, a cell having an intrinsic time of x has a relative progression rate through the cell cycle γ equal to $\frac{1}{t}(x + k \cdot d_{CYC} - d_G)$, with k representing the number of completed cell cycles since the release from growth arrest (based on Equation 1). Hence, considering that γ is normally distributed, we can compute the number of cells in the synchronized population at time t whose cell-intrinsic time is x (denoted $g'(x, t)$) as:

$$g'(x, t) = \sum_{k=0}^2 2^k \cdot e^{-\frac{(x+k \cdot d_{CYC} - d_G - 1)^2}{2\sigma^2}}, \quad (\text{Eq. 2})$$

considering values of k between zero and two, representing three complete cell cycle. We denote by $g(x, t)$ the probability density function of the number of cells at time t whose cell intrinsic time is x , with $g(x, t) = \frac{1}{C} g'(x, t)$, where C is a normalization factor."

7. At the synchronization loss model description, the durations of the cell cycle phases are mentioned ($d_{G1} = 6.8h$, $d_S = 6.4h$ and $d_{G2/M} = 3.2h$). If the G1 and S phases had the same duration, they should be represented equally into the unsynchronized population. However, this is not the case as we can see in Fig. 1A.

Following the reviewer's comment, we realized that the PI staining/FACS analysis of non-synchronized cells was mistakenly performed when cell confluence was too high (potentially out of exponential growth phase) – we thank the reviewer for pointing out this inconsistency. Repeating the experiment with lower confluence cells shows a slightly lower fraction of cells in G1 and higher fraction in S (60% in G1, 26% in S, and 14% in G2/M). Figure 1a was updated accordingly.

To accurately compute the duration of G1, S, and G2/M from these steady-state measurements one needs to consider the fact that at any given time there are substantially more cells with a young age in the population (e.g. the fraction of cells right after mitosis is double that of cells just before mitosis). Specifically, the fraction of cells between time s_1 and s_2 within the cell cycle can be estimated based on the following (e.g., see "A method to estimate cell cycle time and growth fraction using bromodeoxyuridine-flow cytometry data from a single sample", Eidukevicius et al., BMC Cancer 2005):

$$F(s_2) - F(s_1) = 2(\exp(-bs_1) - \exp(-bs_2)), s_1 \leq s_2, b = \frac{\ln(2)}{T_c}$$

where T_c represents the population doubling time. In our case, we get 8.2h for G1, 4.9h for S, and 2.9h for G2/M, versus estimated confidence intervals of the durations of G1, S, and G2/M of 6h-7.6h, 5.9h-6.9h, and 2.8h-3.6h by our analysis of synchronized cells (see Methods). The small under estimation of the duration of G1 and over estimation of the duration of S by the analysis of the synchronized cell (both not more than 1h off the measurements in the non-synchronized cells) may be due to a slight perturbation to cell cycle dynamics due to synchronization-induced growth arrest. We now include the above analysis in the Methods and explicitly relate to a potential perturbation to cell cycle dynamics by the synchronization approach.

8. Right y axis units in Fig. 1A missing.

Fixed.

9. Include also dTTP data in Fig. 2 (from Fig. S11).

We note that Figure 2 shows oscillations in metabolite concentrations while Figure S11b (Figure

S10 in the revised version) shows oscillations in the isotopic labeling of dTTP m+3. Cell cycle oscillations in isotopic labeling of metabolites are shown in Figures 2-3 in the main text, and in several Supp. figures, with all corresponding data included in Supp. Table 2.

10. It is interesting that although pyrimidines peak in the S phase (their synthesis is boosted by glutamate influx into the TCA cycle), the purines do not oscillate in phase. In fact dATP peaks in the S phase, in phase with dTTP and dCTP, but dGTP peaks in the G1 phase. Please comment on this.

While the concentration of dGTP indeed increase in late G1, the m+5 labeling of dGTP upon one-hour feeding with [U-¹³C]-glucose (i.e. labeling of ribose) increases ~2-fold in S phase, suggesting that its biosynthesis increase in S phase (Supp. Table 2). Similarly, the m+5 labeling of GTP and ADP show a marked increase in m+5 labeling in S phase, while the concentration of GTP increase in late G1 and that of ADP do not change significantly throughout the cell cycle. We now refer to the oscillations in m+5 labeling of purines and pyrimidines, peaking in S phase in the Results. This is good example why the cell cycle oscillations in concentrations do not directly represent changes in flux and further highlights the importance of the performed pulse-chase isotope tracing experiments to reveal the metabolic dynamics throughout the cell cycle.

11. Fig. S4: there is not black-dashed line.

Fixed.

12. In the computational deconvolution of metabolite concentration is $u_i(t)$ estimated using the measured or the corrected volume?

We now state that $u_i(t)$ was derived by “normalizing the measured per-cell pool size by the measured average cell volume in time t ”.

13. In Fig. 1C, how was the green line (volume in case of no synchronization loss) determined?

This was determined based on deconvolution of the volume measurements performed on the synchronized cell population, utilizing Equation 6. We now state this in the caption of Figure 1c.

14. Blend figures 3b and S3. In fact, it is confusing when different nonmeclatures are used to describe reactions in the network. Also, with all the labeling patters (e.g. m+2, m+3, m+4, m+5) present all around in the text, it would be nice if those terms were also included in Fig. 3b.

Figure 3b was changed as suggested by the reviewer and Figure S3 removed.

15. Assumptions of the model: Influence of the applied network size? Mitochondrial volume? Metabolites originate from both the cytoplasm and the mitochondrion. What kind of consequences does this have?

We added a discussion of the influence of the network size and subcellular compartmentalization issues in the Discussion (see also reply to comment #2).

16. P. 5. 3rd paragraph: 30% increase in lactate secretion rate. How was lactate measured in the extracellular environment. I might have overlooked it, but I feel that the authors did not mention how they measured extracellular metabolites. Also, how from such extracellular metabolite concentration measurements did they estimates rates?

The extraction procedure for metabolites from the culture media is given in the Methods. Following the reviewers' comments, we now added a detailed description of the computational method used to estimate metabolite secretion throughout the cell cycle via a deconvolution method (see Methods; and also reply to comment #3 by reviewer #2).

17. Method section on LC-based metabolomics: It is stated that conditioned medium was used. Please provide more detail, such that others can in fact reproduce this experiment.

This is now described in the Methods.

18. Page 15: second last line: Grammar?

Fixed.

19. Optimizations were started multiple times, but the authors did not say how often they in fact started the optimizations.

We now indicated that each non-convex optimization problem was run 10 times to potentially avoid local minima.

2nd Editorial Decision

25 September 2017

Thank you again for sending us your revised manuscript. We have now heard back from the two reviewers who were asked to evaluate your manuscript. As you will see below, the reviewers are satisfied with the modifications made. I am therefore pleased to inform you that your paper has been accepted for publication.

REVIEWER REPORTS

Reviewer #1:

The authors have addressed all my questions.

Reviewer #3:

The authors have addressed my previous comments in an excellent manner. This manuscript is now much clearer, and nicer to read. It represents a significant technological advance and provides highly interesting insights. I would like to congratulate the authors on this work, and also would like to thank them for having addressed really all my comments in a perfect manner. Seeing this makes reviewing much more pleasant ...

Corresponding Author Name: Tomer Shlomi

Manuscript Number: MSB-17-7763